# Selective dephosphorylation by PP2A-B55 directs the meiosis I-meiosis II transition in oocytes

S Zachary Swartz[1,2], Hieu T Nguyen[3], Brennan C McEwan[3], Mark E Adamo[4], Iain M Cheeseman[1,2]*, Arminja N Kettenbach[3,4]*

[1]Whitehead Institute for Biomedical Research, Cambridge, United States; [2]Department of Biology, Massachusetts Institute of Technology, Cambridge, United States; [3]Department of Biochemistry and Cell Biology, Geisel School of Medicine at Dartmouth, Hanover, United States; [4]Norris Cotton Cancer Center, Geisel School of Medicine at Dartmouth, Lebanon, United States

**Abstract** Meiosis is a specialized cell cycle that requires sequential changes to the cell division machinery to facilitate changing functions. To define the mechanisms that enable the oocyte-to-embryo transition, we performed time-course proteomics in synchronized sea star oocytes from prophase I through the first embryonic cleavage. Although we found that protein levels were broadly stable, our analysis reveals that dynamic waves of phosphorylation underlie each meiotic stage. We found that the phosphatase PP2A-B55 is reactivated at the meiosis I/meiosis II (MI/MII) transition, resulting in the preferential dephosphorylation of threonine residues. Selective dephosphorylation is critical for directing the MI/MII transition as altering PP2A-B55 substrate preferences disrupts key cell cycle events after MI. In addition, threonine to serine substitution of a conserved phosphorylation site in the substrate INCENP prevents its relocalization at anaphase I. Thus, through its inherent phospho-threonine preference, PP2A-B55 imposes specific phosphoregulated behaviors that distinguish the two meiotic divisions.

*For correspondence:
icheese@wi.mit.edu (IMC);
Arminja.N.Kettenbach@
dartmouth.edu (ANK)

Competing interests: The authors declare that no competing interests exist.

## Introduction

Animal reproduction requires that oocytes undergo a specialized cell cycle called meiosis, in which two functionally distinct divisions occur in rapid succession to reduce genome ploidy (*Kishimoto, 2018*). Following fertilization, the oocyte must then transition to a third division strategy, mitosis, for early embryonic development. This oocyte-to-embryo transition occurs in a short temporal window, but must achieve high fidelity to ensure that heritable information is accurately transmitted from the parents to the developing embryo. At the center of this progression is a suite of cell cycle regulatory proteins and molecular machines that drive and integrate processes such as chromosome segregation, fertilization, and pronuclear fusion. An important goal is to unravel the complex regulatory mechanisms that precisely coordinate these divisions in time and space within the oocyte. In particular, it remains unknown how phosphorylation and dephosphorylation drive the meiotic divisions allowing oocytes to rewire the cell division machinery at the meiosis I/meiosis II (MI/MII) transition to facilitate differing requirements.

The female meiotic cell cycle is distinct from mitosis in several ways, necessitating a unique regulatory control. First, oocytes remain in an extended primary arrest in a cell cycle state termed prophase I until receiving an extrinsic hormonal signal (*Conti and Chang, 2016*; *Jaffe and Norris, 2010*; *Kishimoto, 2018*; *Von Stetina and Orr-Weaver, 2011*). Second, the meiotic divisions use a small asymmetrically positioned spindle to partition chromosomes into polar bodies, which do not contribute to the developing embryo (*Severson et al., 2016*). In addition, the first meiotic division

segregates bivalent pairs of homologous chromosomes, whereas for MII, this configuration is reversed and instead sister chromatids are segregated (*Figure 1A*; *Watanabe, 2012*). Finally, meiosis lacks a DNA replication phase between the polar body divisions, which enables the reduction of ploidy to haploid. How the cell division machinery is specialized to perform the distinct functions of MI, and then is rapidly reorganized for the unique requirements of MII while remaining in meiosis and not exiting into gap or S-phase, is an important open question.

In this study, we define phosphoregulatory mechanisms that drive the MI/MII transition. We undertook a proteomic and phosphoproteomic strategy using oocytes of the sea star *Patiria miniata*, which undergoes meiosis with high synchrony (*Swartz et al., 2019*). Prior analyses have revealed proteome-wide changes in animal models including *Xenopus*, *Drosophila*, and sea urchins (*Guo et al., 2015*; *Krauchunas et al., 2012*; *Presler et al., 2017*; *Zhang et al., 2019*). However, the biology of these organisms limits access to a comprehensive series of time points spanning prophase I through the embryonic divisions, including the critical MI/MII transition. Our sea star proteomics dataset spans the entire developmental window from prophase I arrest through both meiotic divisions, fertilization, and the first embryonic division (*Figure 1A*). We identified a surprising differential behavior between serine and threonine dephosphorylation at the MI/MII transition that we propose to underlie key regulatory differences between these meiotic divisions. This regulatory switch is driven by PP2A-B55, which is reactivated after MI to preferentially dephosphorylate threonine residues, thereby creating temporally distinct reversals of cyclin-dependent kinase (CDK) and mitogen-activated protein kinase (MAPK) phosphorylation. We propose a model in which the usage of threonine vs serine endows substrates with different responsivity to a common set of kinases and phosphatases, temporally coordinating individual proteins with meiotic cell cycle progression to achieve specific behaviors for MI and MII without exiting from meiosis.

## Results

### Proteomic analysis reveals stable protein abundance during the oocyte-to-embryo transition

The oocyte-to-embryo transition involves an ordered series of events including fertilization, chromosome segregation, polarization, and cortical remodeling. To determine the basis for these cellular transitions and their corresponding physical changes, we analyzed the proteome during the oocyte-to-embryo transition using quantitative tandem mass tag (TMT)-multiplexed mass spectrometry (MS) (*Thompson et al., 2003*). We obtained prophase I-arrested oocytes from the sea star *P. miniata* and treated them with the maturation-inducing substance 1-methyladenine (1-MeAd) to trigger the resumption of meiotic progression in seawater culture (*Kanatani et al., 1969*). In an initial trial, we collected oocytes for immunofluorescence every 10 min after 1-MeAd stimulation and determined that oocytes proceed through meiosis with high synchrony (over 90% synchrony of oocytes at germinal vesicle breakdown (GVBD), MI, MII, and pronuclear stages; *Figure 1—figure supplement 1*). Live-imaging experiments further supported synchronous progression through meiosis and early development in this species (*Figure 1—video 1*). Leveraging these features, we cultured isolated oocytes and collected biological triplicate samples at the following stages: (1) prophase I arrest (Pro), (2) GVBD, (3) metaphase of MI, (4) prometaphase of MII, (5) just prior to pronuclear fusion (2-PN), and (6) metaphase of the first embryonic cleavage (FC) (*Figure 1A*).

We first tested whether protein abundance changes could regulate the oocyte-to-embryo transition (*Figure 1B*). Using MS, we identified 8026 total proteins, of which 6212 were identified in two independent time-course series and 4635 in all three series (*Figure 1—figure supplement 2*, *supplementary file 1*). Surprisingly, only a limited number of proteins changed in abundance across these different stages (*Figure 1C*). In fact, 99% of the 4635 proteins reproducibly identified in all three time-course series displayed a maximum fold change of less than 2 from prophase I to the first embryonic cleavage, with 74.8% of proteins displaying less than a 1.2-fold change (*Figure 1D*). The absence of changes in protein levels was not due to a bias in our analysis as protein abundance followed a normal distribution (*Figure 1E*). We also note that this limited protein turnover during meiosis is consistent with prior reports in other organisms (*Kishimoto, 2018*; *Peuchen et al., 2017*; *Presler et al., 2017*). However, despite this broad stability of protein levels, there were several notable exceptions (*Figure 1F*). Gene ontology (GO) analysis (*Liao et al., 2019*) of significantly regulated

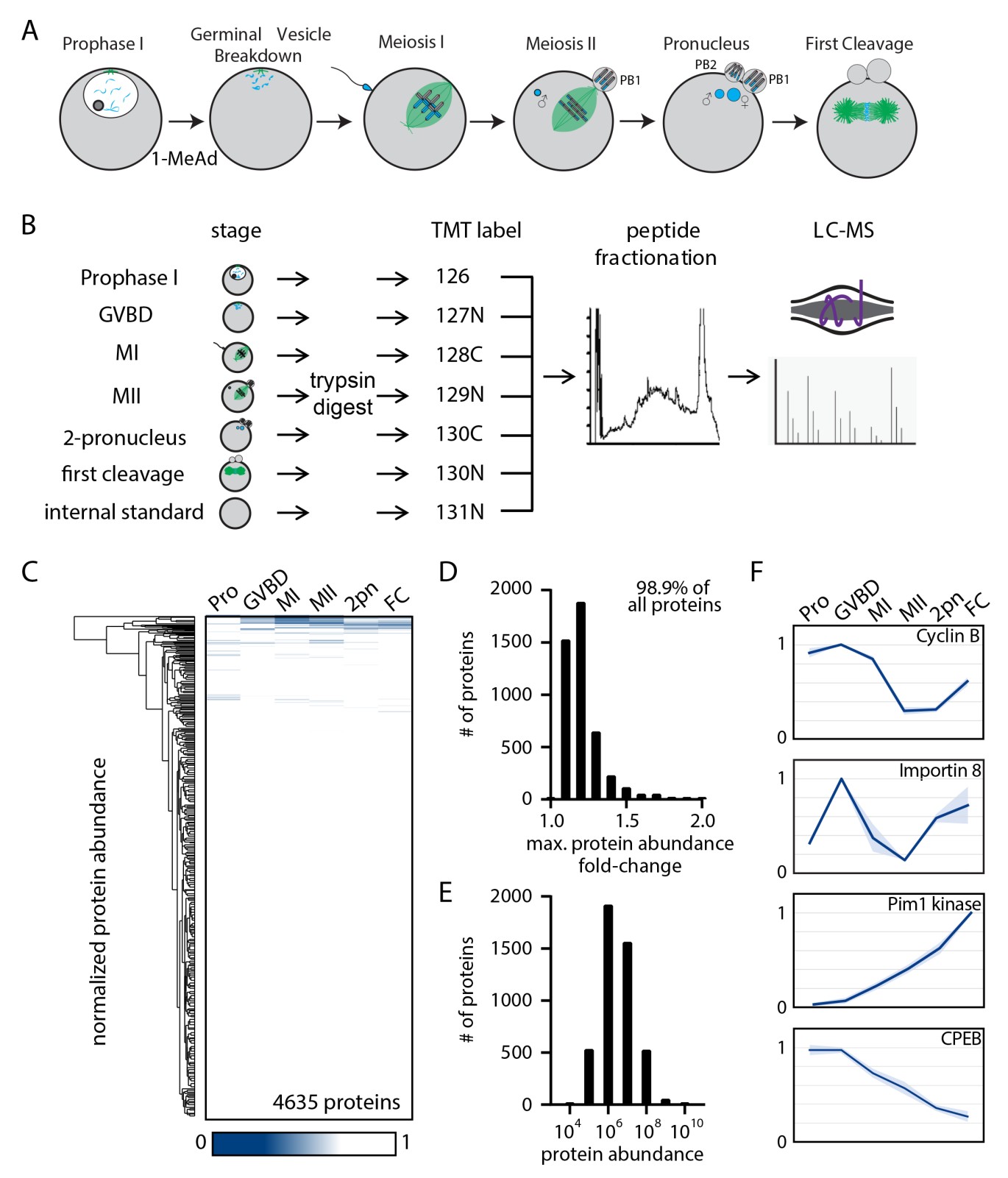

**Figure 1.** The proteome during the oocyte-to-embryo transition is broadly stable. (**A**) Schematic of meiotic progression in sea star oocytes, representing the six stages collected for mass spectrometry analysis. (**B**) Proteomics workflow diagram, in which protein samples were collected in biological triplicates, digested, tandem mass tag (TMT) labeled, fractionated, and analyzed by liquid chromatography with mass spectrometry (LC-MS). (**C**) Hierarchical clustering of the relative abundance of 4635 proteins detected across three replicates. Individual proteins are clustered (vertically) by

*Figure 1 continued on next page*

*Figure 1 continued*

the six isolated meiotic stages (horizontally). (D) Histogram of proteins binned by their maximum fold change in abundance, indicating 98.99% of all proteins undergo a fold change of less than 2. (E) Abundance histogram of proteins identified in our analysis reveals a normal distribution. (F) Relative abundances of selected proteins across stages (Pro, prophase I; GVBD, germinal vesicle breakdown; MI, meiosis I; MII, meiosis II; 2-PN, two pronucleus; FC, first cleavage). Light blue shading represents standard deviation.

The online version of this article includes the following video and figure supplement(s) for figure 1:

**Figure supplement 1.** Oocyte synchrony analysis.

**Figure supplement 2.** Mass spectrometry analysis of protein abundance.

**Figure 1—video 1.** Time-lapse imaging of sea star oocytes progressing through meiosis.

https://elifesciences.org/articles/70588#fig1video1

proteins with a fold change of 1.2 or more revealed an enrichment in cytoskeletal proteins, proteins involved in RNA binding, and ribosomal components (*Supplementary file 2*). Moreover, cyclin B levels were high in prophase I, GVBD, and MI, but declined sharply in MII, before being partially restored in the first cleavage stage (*Figure 1F*). These dynamics are consistent with Anaphase Promoting Complex/Cyclosome (APC/C) mediated destruction of cyclin B during cell cycle progression (*Evans et al., 1983*; *Kishimoto, 2018*; *Okano-Uchida et al., 1998*). In addition, we identified the serine/threonine kinase Proviral integration site for Moloney murine leukemia virus-1 (PIM-1) (*Figure 1F*), previously proposed to be a potential mitotic regulator (*Bachmann et al., 2006*), as a protein that is absent in prophase I oocytes but translated de novo following meiotic resumption. Thus, although the overall proteome is highly stable, selected proteins are translationally or proteolytically regulated with meiotic cell cycle progression.

## New translation of selected proteins is required for meiotic progression

Although protein levels are largely constant across the oocyte-to-embryo transition, de novo translation could act to maintain steady-state levels or may be required to produce a limited set of factors involved in meiotic progression. To test this, we globally prevented translation using the 40S ribosomal inhibitor emetine (*Jimenez et al., 1977*). Emetine-treated oocytes responded to 1-MeAd stimulation to initiate MI, consistent with prior work (*Houk and Epel, 1974*). However, instead of progressing to MII, the maternal DNA decondensed and formed a pronucleus with oocytes remaining arrested even at time points when control oocytes had initiated first cleavage (*Figure 2A,B*). Based on a proteomic analysis of MII and pronuclear stage oocytes, we found that only 108 out of 7610 proteins identified in our analysis significantly changed with emetine treatment (*Figure 2—figure supplement 1*, *supplementary file 3*). These emetine-sensitive proteins fell into diverse categories but were overrepresented for cytoskeletal elements and actomyosin-related proteins (*Supplementary file 4*). Notably, we identified the DNA replication factor Cdt1, whose nascent translation may be required for the initiation of DNA replication after meiotic exit. Moreover, we again identified PIM-1 as a factor sensitive to emetine treatment, consistent with our steady-state analysis (*Figure 1F*). In summary, most proteins are insensitive to translational inhibition, indicating a general lack of turnover between MI and MII, but new protein synthesis is required for progression past MI.

The requirement of protein synthesis for meiotic progression could reflect the need to translate selected cell cycle factors. To test whether established cell cycle regulators must be translated de novo, we used morpholino injection to specifically prevent new translation of cyclin B, one of the few proteins that varies in abundance (*Figure 2C,D*), as well as cyclin A, which is synthesized in late MI in the related sea star species *Patiria pectinifera* (*Hara et al., 2009*; *Okano-Uchida et al., 1998*). Morpholino injection specifically reduced cyclin B protein levels and reduced total CDK-consensus phosphorylation (*Figure 2—figure supplement 2*). We stimulated oocytes with 1-MeAd immediately following morpholino injection to ensure that pre-existing cyclin protein was unaffected. When new cyclin A synthesis was blocked, oocytes underwent both meiotic divisions normally and the maternal and paternal pronuclei fused, but these zygotes then arrested with a single fused pronucleus and failed to progress to the first cleavage (*Figure 2C,D*). This is consistent with a role for cyclin A in mitotic entry in cultured cells, and the transition to embryogenesis in *P. pectinifera* (*Hara et al., 2009*; *Okano-Uchida et al., 2003*; *Pagano et al., 1992*). In contrast, preventing cyclin B synthesis

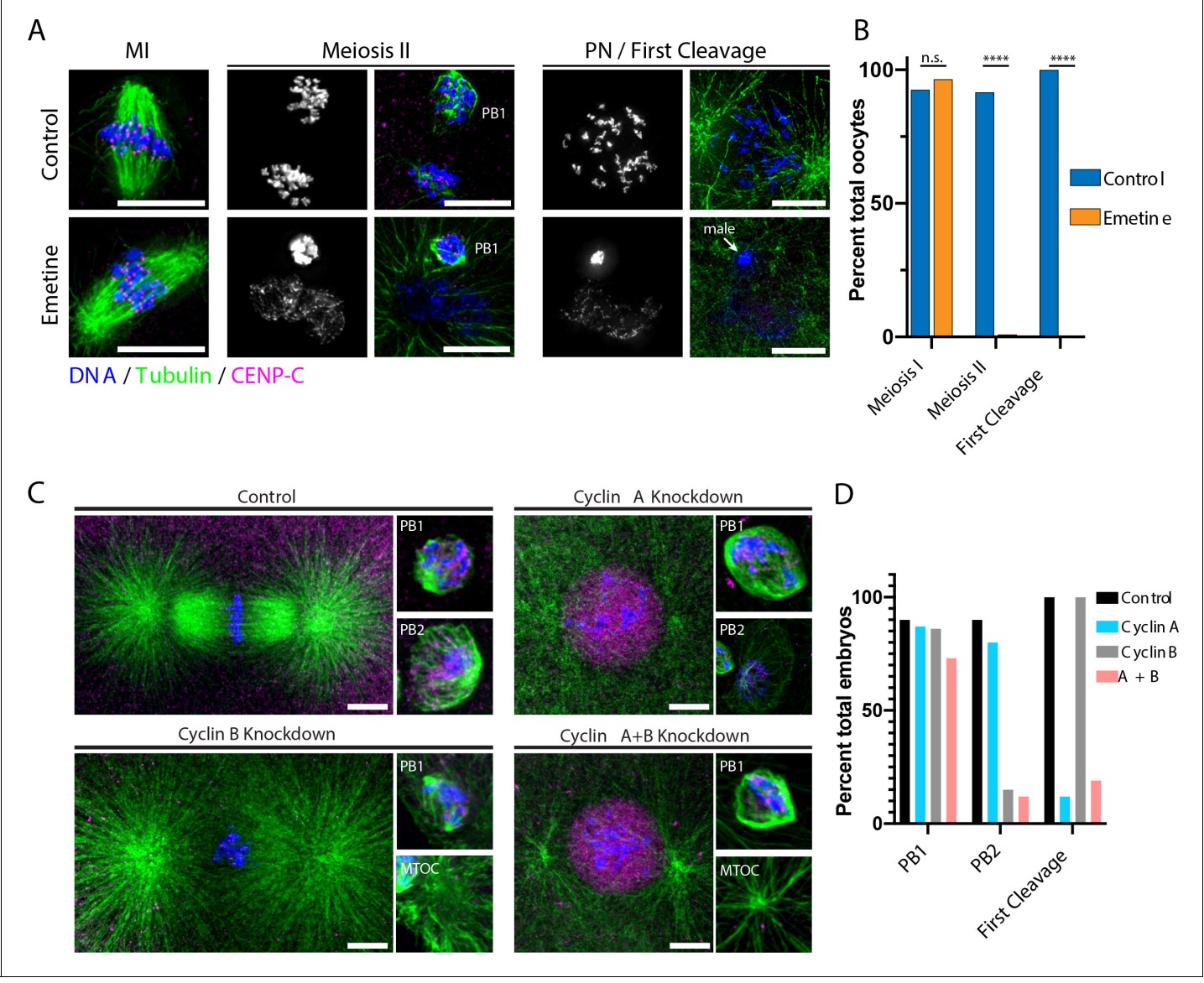

**Figure 2.** Protein synthesis is required for the MI/MII transition. (**A**) Immunofluorescence of control or emetine-treated oocytes, with DNA provided in single-channel grayscale images. While controls proceed to meiosis II (MII) and first cleavage, emetine-treated oocytes decondense DNA after meiosis I (MI), arrest in a pronuclear state, and fail to incorporate the male DNA. Microtubules were scaled nonlinearly. Scale bars = 10 μm. (**B**) Fraction of oocytes that successfully extruded both polar bodies and underwent first cleavage (MI: control n = 107, emetine n = 113; MII: control n = 107, emetine n = 114; first cleavage: control n = 31, emetine n = 47 oocytes; ****p<0.0001 by Fisher's exact test). (**C**) Immunofluorescence of oocytes in which nascent synthesis of cyclin A, cyclin B, or both was blocked. Control oocytes extruded both polar bodies and initiated first cleavage. Blocking cyclin A synthesis did not affect the meiotic divisions but caused an arrest prior to the first cleavage. Blocking cyclin B instead selectively disrupted the second mitotic division, but the first meiosis and initiation of first cleavage proceeded normally. Combined translational inhibition of both cyclin A and cyclin B resulted in an interphase arrest following the first meiotic division. Microtubules were scaled nonlinearly. Scale bars = 10 μm. (**D**) Fraction of oocytes that successfully extruded polar bodies and underwent first cleavage (cyclin A n = 82, cyclin B n = 66, cyclin A + B n = 52 oocytes).

The online version of this article includes the following source data and figure supplement(s) for figure 2:

**Figure supplement 1.** Mass spectrometry analysis of nascent protein synthesis.

**Figure supplement 2.** Specificity of cyclin morpholino knockdowns.

**Figure supplement 2—source data 1.** Western blots following cyclin morpholino knockdown.

resulted in a normal MI division but failure to extrude the second polar body and retention of an additional centriole. Surprisingly, these oocytes successfully underwent pronuclear fusion, entered the first cleavage, and formed a mitotic spindle. This suggests that cyclin B must be translated de novo following anaphase I to drive meiosis II but is dispensable for the initial transition from meiosis to embryonic mitosis. Finally, when we simultaneously prevented the new translation of both cyclin A and B, oocytes completed MI and then arrested in a pronuclear-like state without conducting MII (*Figure 2C,D*), similar to the effect of translational inhibition by emetine (*Figure 2A,B*). We further attempted to prevent translation of PIM-1 but did not observe a substantial phenotype, which could reflect technical challenges in its knockdown. Taken together, our results suggest that the proteome during the oocyte-to-embryo transition is highly stable but that the de novo translation of cyclins is required for meiotic progression.

## Defining the phosphorylation landscape of the oocyte-to-embryo transition

The two meiotic divisions, fertilization, pronuclear fusion, and the first mitotic cleavage all occur within less than 3 hr in the absence of substantial changes in protein abundance (*Figure 1C,D*). This suggests that there are alternative mechanisms to rapidly re-organize the cell division apparatus during these transitions. Therefore, we next assessed phosphorylation across the oocyte-to-embryo transition using phosphopeptide enrichment followed by MS (*Figure 3A*). Our analysis identified a total of 25,228 phosphopeptides across three multiplexed time courses. Among those phosphopeptides, 16,691 were identified in two, and 11,430 in three, multiplexes (*Figure 3—figure supplement 1A*, *supplementary file 5*). We detected 79.3% of phosphorylation on serine residues, 19.6% on threonine, and 1.1% on tyrosine residues based on a phosphorylation localization probability of $\geq 0.9$ (*Figure 3—figure supplement 1B*). Prior work found similar ratios of S:T:Y phosphorylation based on autoradiographic measurements in chicken cells (92%:7.7%:0.3%) and based on phosphoproteomics in human cells (84.1%:15.5%:0.4%) (*Hunter and Sefton, 1980*; *Sharma et al., 2014*).

Hierarchical clustering of the dynamic phosphorylation behavior from prophase I to the first embryonic cleavage revealed several striking transitions in global phosphorylation status (*Figure 3B*). First, prophase I-arrested oocytes are distinct from those in the other stages in that they not only display a limited number of phosphorylation sites at the relative maximum phosphorylation levels (*Figure 3C*) but also have the lowest overall phosphorylation state of the samples tested (*Figure 3D*). Second, more than half of the total phosphorylation sites identified were maximally phosphorylated in MI, whereas phosphorylation was substantially reduced in MII (*Figure 3C*). These patterns of phosphorylation imply a critical role for phosphoregulation in specializing the two meiotic and first cleavage divisions and suggest a role for a low phosphorylation state in maintaining the prophase I arrest.

## Kinase activity across the oocyte-to-embryo transition

To remain arrested in prophase I, CDKs must be kept inactive. As our proteomics analysis indicated that the majority of proteins in the oocytes, including kinases, are present constitutively (*Figure 1C*), kinase activity must be controlled post-translationally. We, therefore, analyzed the pattern of phosphorylation events on established cell cycle kinases. We first identified inhibitory phosphorylation sites on Cdk1 or Cdk2 (Y21 or Y15, respectively) by proteomics and western blotting using phospho-specific antibodies against these conserved sites (*Figure 3E,F*). These sites are phosphorylated in prophase I arrest, and to a lesser extent at first cleavage, but not during meiosis. This phosphorylation pattern suggests that Cdk is inactive in prophase I-arrested oocytes but active throughout the meiotic divisions. In addition to Cdk, the meiotic divisions and secondary arrest that occurs in the absence of fertilization require MAP kinase activity downstream of the conserved activator Mos (*Dupré et al., 2011*; *Tachibana et al., 2000*). Based on our phosphoproteomics and western blotting, we found that a conserved activating phosphorylation on the MAP kinase p42/ERK phosphorylation (Y204) was undetectable in prophase I-arrested oocytes, high in MI and MII, and low in first cleavage (*Figure 3G,H*).

Meiotic resumption in sea star oocytes requires the action of kinases downstream of the 1-MeAd hormone G protein-coupled receptor (GPCR) on the oocyte surface (*Chiba et al., 1992*). Hormonal reception leads to activation of phosphoinositide 3-kinase (PI3K), which in turn phosphorylates

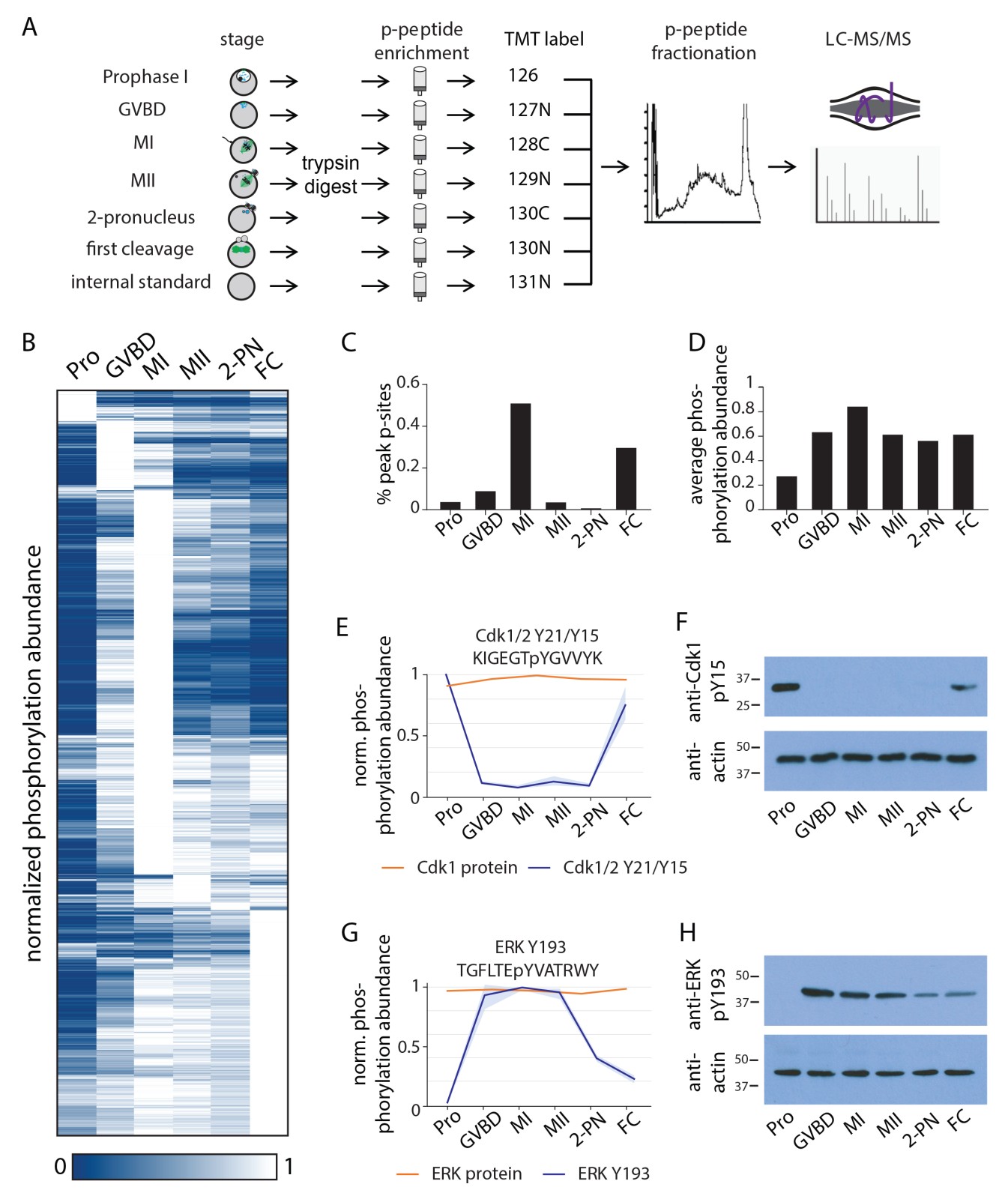

**Figure 3.** Identification of dynamic phosphorylation changes during the oocyte-to-embryo transition. (**A**) Proteomics workflow diagram, in which a phosphopeptide enrichment step was performed prior to tandem mass tag (TMT) labeling. (**B**) Hierarchical clustering of 10,645 phosphorylation events (a localization score of 0.9 or higher, a p-value of 0.05 or less, and stage-specific peak phosphorylation) clustered by phosphosite (rows) and meiotic stage (columns). (**C**) Percentage of sites that reach their peak phosphorylation levels across stages, revealing peaks at meiosis I (MI) and first cleavage.

*Figure 3 continued on next page*

*Figure 3 continued*

(D) Average abundance of all phosphorylation events per stage. (E, G) Temporal phosphorylation levels of conserved sites on the kinase Cdk1/2 and the MAP kinase Erk (blue trace, light blue areas are standard deviations; orange trace represents total relative protein abundance). (H, F) Western blots with antibodies recognizing the inhibitory phosphorylation on Cdk1Y15 and activating phosphorylation on ERK Y193, respectively.

The online version of this article includes the following source data and figure supplement(s) for figure 3:

**Source data 1.** Identification of dynamic phosphorylation changes during the oocyte-to-embryo transition.
**Figure supplement 1.** Mass spectrometry analysis of phosphorylation abundance.
**Figure supplement 2.** Phosphorylation behavior of cell cycle regulators.
**Figure supplement 3.** RVp[S/T]F motif phosphorylation.
**Figure supplement 3—source data 1.** RVp[S/T]F western blots.
**Figure supplement 4.** NIPP1 activity in arrest and meiosis.
**Figure supplement 4—source data 1.** NIPP1 and PP1 western blots.
**Figure supplement 5.** ARPP19 activity in arrest and meiosis.
**Figure supplement 5—source data 1.** ARPP19 activity in arrest and meiosis.
**Figure supplement 6.** Phosphoprotein phosphatase function in arrested oocytes and meiosis.
**Figure supplement 7.** Effect of phosphatase inhibition on phosphorylation, meiotic arrest, and resumption.

and activates the meiotic trigger kinase, which has been previously proposed to be RAC-alpha protein kinase (AKT) kinase (*Okumura et al., 2002*). However, recent work has instead suggested that SGK kinase is the primary trigger in sea star *P. pectinifera,* which also acts downstream of PI3K (*Hiraoka et al., 2019*). In support of this model, we detected an increase in phosphorylation of the SGK activation segment following hormonal stimulation (*Figure 3—figure supplement 2A*). SGK then phosphorylates the CDK regulators Cdc25 and Myt1 to activate and repress them, respectively, thereby enabling the activation of Cdk1 (*Hiraoka et al., 2019*). Concordantly, we observed an increase in phosphorylation on Cdc25 S188, an activating SGK site, and Myt1 S75, a repressive SGK site, following hormonal stimulation (*Figure 3—figure supplement 2B,C*; *Hiraoka et al., 2019*). Finally, to assess the relative contributions of AKT and SGK to meiotic resumption, we investigated the presence of RxRxxS/T consensus motifs in the dataset for the localized, single, and reproducibly quantified phosphopeptides. Based on known AKT and SGK substrates, the AKT consensus motif shows enrichment in basophilic amino acids in the −4 position compared to SGK (*Hornbeck et al., 2015*). We then compared the sites that increased in phosphorylation abundance by threefold or more from prophase I to GVBD to sites with less than a threefold increase over the same time period. The sites that increased in abundance by more than threefold were enriched for a SGK-like consensus motif, whereas those that increased less displayed a more AKT-like motif (basophilic residues in the −4 position) (*Figure 3—figure supplement 2D*). Collectively, these results support a model where SGK acts as the major trigger kinase driving meiotic resumption (*Hiraoka et al., 2019*).

Finally, meiotic progression also requires Greatwall kinase (*Kishimoto, 2018*), which acts upstream to inhibit PP2A-B55. Greatwall is sequestered in the germinal vesicle in prophase I sea star oocytes and is activated downstream of Cdk1/cyclin B (*Hara et al., 2012*). We identified a conserved activating phosphorylation within the activation segment of Greatwall kinase (T194 in humans; T204 in sea star) (*Figure 3—figure supplement 2E*; *Blake-Hodek et al., 2012*; *Gharbi-Ayachi et al., 2010*), indicative of high Greatwall kinase activity during GVBD and MI, and reduced activity in later stages. In summary, our phosphoproteomic time course reveals orchestrated transitions in the activity of regulatory kinases during the oocyte-to-embryo transition.

## Prophase I arrest is enforced by high phosphatase activity

Our phosphoproteomics analysis indicated that prophase I is characterized by low global phosphorylation. To determine whether this state is reinforced by phosphatase activity, we next examined modifications to the major cell cycle phosphatases, Protein Phosphatase 1 (PP1) and Protein Phosphatase 2A (PP2A), and their regulators (*Nasa and Kettenbach, 2018*). PP1 activity is inhibited by phosphorylation of its catalytic subunits (phosphorylation of T320 (human) by Cdk1) (*Dohadwala et al., 1994*; *Kwon et al., 1997*). PP1 T316 (corresponding to T320 in humans) is hypophosphorylated in prophase I-arrested oocytes (*Figure 3—figure supplement 2F*), which would result in high PP1 activity. PP1 activity is also controlled through regulatory subunits, which it recognizes through short-linear motifs, most prominently the 'RVxF' motif (*Bollen et al., 2010*;

*Heroes et al., 2013*). Phosphorylation of the 'x' position within an RVxF motif or of adjacent residues disrupts the interaction between regulatory and catalytic subunits (*Nasa et al., 2018*). Analysis of the phosphorylation abundance of RVxF motifs across the phosphoproteomics time course revealed low RVxF phosphorylation occupancy in prophase I, which increased in GVBD and MI, before decreasing in later stages (*Figure 3—figure supplement 2G*). To validate this observation, we performed western blots using an antibody raised against a phosphorylated RVp[S/T]F epitope (*Nasa et al., 2018*) and observed similar trends (*Figure 3—figure supplement 3*). This suggests that, in prophase I, the PP1 catalytic subunit is maximally bound to regulatory subunits to promote substrate dephosphorylation.

PP1-interacting proteins can act as targeting subunits, but can also act as PP1 inhibitors, resulting in a complex PP1 regulatory network (*Bollen et al., 2010*; *Heroes et al., 2013*; *O'Connell et al., 2012*). For example, the association of PP1 with the regulatory factor NIPP1 (nuclear inhibitor of PP1) inhibits its activity but can also recruit some substrates for dephosphorylation. NIPP1 association with PP1 is negatively regulated by phosphorylation of an RVxF motif at the binding interface (*O'Connell et al., 2012*). Indeed, we found that phosphorylation adjacent to its RVxF is low in prophase I-arrested oocytes, but increases following meiotic resumption (*Figure 3—figure supplement 4A–C*). Preventing NIPP1 phosphorylation with phospho-inhibitory mutations (*P. miniata* NIPP1 S199A or S197A S199A double mutants) resulted in a greater than fivefold increase in the binding of sea star NIPP1 to human PP1 when expressed in human 293T cells (*Figure 3—figure supplement 4D*; *Beullens et al., 1992*; *Beullens et al., 1999*; *Tanuma et al., 2008*; *Winkler et al., 2015*). Thus, the low NIPP1 phosphorylation in prophase I oocytes would increase the association between PP1 and NIPP1, modulating and reducing its phosphatase activity toward specific substrates. Taken together, these observations indicate that there is complex regulation of PP1 activity in prophase I involving multiple activating and inhibitory holoenzyme complexes.

We also identified phosphorylation events predicted to regulate PP2A activity through its B55 subunit (*Figure 3—figure supplement 5A*). Upon phosphorylation of ARPP19 by Greatwall kinase (on S67 in humans, S106 in *P. miniata*), phospho-ARPP19 binds to and inactivates the regulatory PP2A regulatory subunit B55 (*Gharbi-Ayachi et al., 2010*; *Okumura et al., 2014*). Indeed, we found that PmARPP19 thio-phosphorylated in vitro by Greatwall kinase resulted in increased Cdc25 phosphorylation when added to mitotic human cell lysates (*Figure 3—figure supplement 5B*). Thus, phospho-ARPP19 S106 acts as an active B55 inhibitor such that ARPP19 phosphorylation serves as a proxy for assessing PP2A-B55 activity in sea star oocytes. PmARPP19 S106 phosphorylation is low in prophase I, but is high in GVBD and MI, before decreasing again in MII (*Figure 3—figure supplement 5C*). ARPP19 is additionally activated by Cdk1 phosphorylation on S69 (*Figure 3—figure supplement 5C*; *Okumura et al., 2014*). Consistent with this regulation, we observed increasing phosphorylation on S69 following meiotic resumption (*Figure 3—figure supplement 5C*). Based on the collective behavior of these regulatory phosphorylation events on ARPP19, we conclude that PP2A-B55 activity is high in prophase I, low in MI, but is reactivated at the MI/MII transition.

To test the functional requirement for these phosphatases in maintaining the prophase I arrest, we treated oocytes with the potent dual PP1 and PP2A inhibitor calyculin A. Following calyculin A addition, we found that 100% of oocytes spontaneously underwent GVBD within 70 min (*Figure 3—figure supplement 6A,C*; also see *Tosuji et al., 1991*). To determine whether this effect was due to inhibition of PP1 or PP2A, we used the selective PP1 inhibitor tautomycetin (*Choy et al., 2017*; *Swingle et al., 2007*). Using in vitro phosphatase assays, we confirmed that tautomycetin potently inhibited PP1 (*Figure 3—figure supplement 6B*). In contrast to calyculin A treatment, tautomycetin did not induce GVBD across a wide range of concentrations (*Figure 3—figure supplement 6C*). However, following meiotic resumption induced by hormonal stimulation, tautomycetin treatment did cause meiotic chromosome alignment and segregation errors (*Figure 3—figure supplement 6D*). This phenotype supports a role for PP1 in regulating kinetochore-microtubule attachments and other activities during meiosis. We therefore conclude that PP1 is dispensable for maintaining a prophase I arrest but is required to achieve high fidelity chromosome segregation during meiosis, with PP2A playing the critical role in maintaining the prophase I arrest.

To test the contributions of PP1 and PP2A, we next conducted a phosphoproteomic analysis of calyculin A-treated oocytes. This analysis revealed a broad upregulation of phosphorylation (*Supplementary file 5*), with 85% of phosphorylation sites displaying a high level of phosphorylation after 70 min of calyculin A treatment (*Figure 3—figure supplement 7A,B*). However,

phosphorylation sites with a normal maximal phosphorylation occupancy in prophase I (when PP2A displays high activity) decreased after 70 min of calyculin A treatment (*Figure 3—figure supplement 7B*). Although global phosphorylation levels increased similarly to those observed upon meiotic entry, calyculin A-treated oocytes failed to progress past this GVBD-like state based on the absence of a contractile actin network, chromosome congression, or spindle formation (*Figure 3—figure supplement 7C*). Therefore, although PP2A phosphatase activity is required to maintain a normal prophase I arrest, its inhibition is not sufficient to recapitulate physiological meiotic resumption. Collectively, our phosphoproteomic and functional analyses reveal high PP2A activity in prophase I-arrested oocytes, which enables a low global phosphorylation state, with additional waves of PP1 and PP2A phosphatase activity controlling subsequent phases of meiotic phosphorylation.

## PP2A-B55 drives selective dephosphorylation at the MI/MII transition

During meiosis, oocytes must undergo two consecutive chromosome segregation events without exiting into an interphase state. These rapid divisions occur within 30 min of each other in the sea star, but they each achieve distinct functions. Therefore, a subset of MI-associated phosphorylation events must be reversed to allow progression to MII, whereas others must be maintained to remain in meiosis. Of the stages tested, MI displayed the largest number of sites with maximal phosphorylation (*Figure 3A*), including a large number of sites with a TP or SP consensus motif indicative of CDK- or MAPK-dependent phosphorylation (*Supplementary file 5*). Moreover, phosphorylation of these sites increased when oocytes were treated with the calyculin A, indicating that they are putative substrates for PP1 or PP2A (*Figure 3—figure supplement 7A,B*, *supplementary file 5*). Hierarchical clustering of peptides with a single phosphorylation site maximally phosphorylated in MI revealed three distinct clusters (*Figure 4A*). A subset of sites sharply decreased in their phosphorylation after MI (*Figure 4B*, cluster 3), whereas other sites remained phosphorylated during MII and the first embryonic cleavage (*Figure 4B*, cluster 2). A third cluster displayed intermediate dephosphorylation kinetics (*Figure 4B*, cluster 1). These differential behaviors could provide a mechanism by which the meiotic divisions are specified and underlie the transition from MI to MII.

To determine the mechanisms controlling these differential phosphorylation behaviors, we analyzed the specific phosphorylation events that are eliminated after MI (*Figure 4B*, cluster 3). Motif analysis of single phosphorylation sites in cluster 3 revealed that they predominantly occur on threonine with proline in the +1 position (TP sites). Indeed, by directly comparing the phosphorylation of SP vs TP sites in the three clusters, we found that TP phosphorylation declined more substantially than SP after MI (*Figure 4D*). On average, the mean phosphorylation of TP motifs detected by phosphoproteomics peaked in MI, but then declined substantially in MII. In contrast, SP motif phosphorylation decreased to a lesser extent (*Figure 4F*). To evaluate differential SP vs TP phosphorylation at other cell cycle stages, we additionally analyzed their relative abundances in prophase I-arrested oocytes and found that TP phosphorylation is lower on average (*Figure 4—figure supplement 1B*). Thus, our analysis reveals an unanticipated difference in the phosphorylation behavior of serine- vs threonine-containing sites at the MI/MII transition.

The behavior of these sites in our proteomics analysis suggests TP sites are selectively dephosphorylated between MI and MII, whereas SP sites remain phosphorylated. However, an alternative interpretation is that both SP and TP sites are dephosphorylated equally at the MI/MII transition, but then SP sites are selectively and rapidly re-phosphorylated in MII. To distinguish between these models, we collected samples with an increased temporal resolution (every 10 min) throughout meiosis and performed western blots using antibodies against phosphorylated pTP and pSP CDK motifs. Consistent with our MS results, overall TP site phosphorylation peaked in MI, followed by strong reduction in MII. In contrast, SP phosphorylation declined less substantially at the MI/MII transition (*Figure 4G*, *Figure 4—figure supplement 1C*). The same trends were observed with antibodies recognizing TPP and SPP phosphorylation motifs (*Figure 4—figure supplement 2A,B*). Thus, despite the transient inactivation of Cdk1 between the meiotic divisions (*Okano-Uchida et al., 1998*), SP phosphorylation remains relatively stable during this window. We note that MAPK activity remains high between both meiotic divisions (*Figure 3G,H*) and could contribute to the relative stability of SP phosphorylation events during this window. However, this model would still require a phosphatase activity to preferentially oppose TP phosphorylation sites to account for the difference in SP vs TP behavior. Taken together, these data reveal unexpected differences in SP vs TP phosphorylation

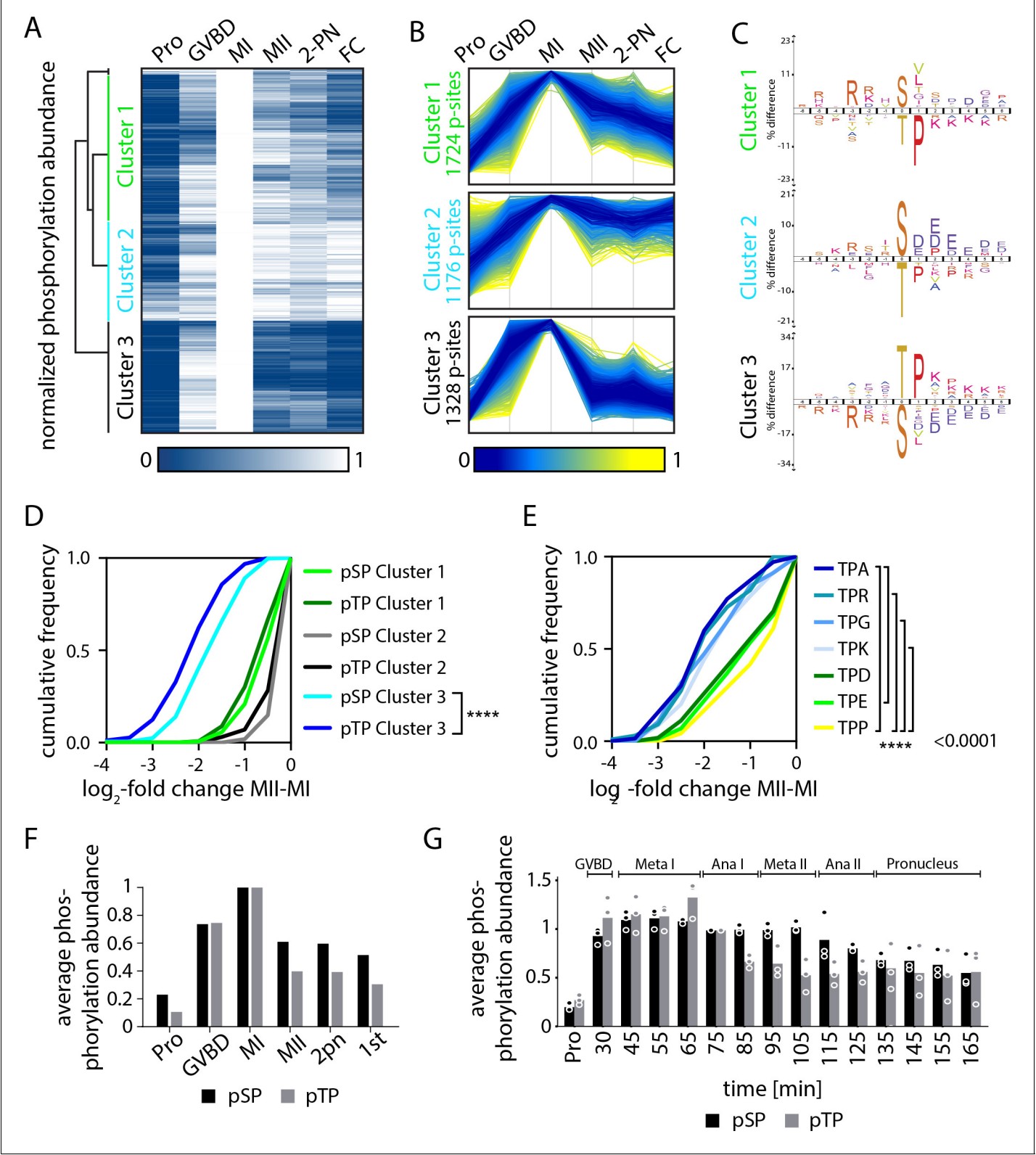

**Figure 4.** Serine and threonine display distinct phosphorylation behaviors. (**A**) Heatmap representation of a subset of sites that peak in phosphorylation in meiosis I (MI). Hierarchical clustering identifies three phosphorylation clusters with distinct temporal behaviors, indicated by green (cluster 1), cyan (cluster 2), and black (cluster 3) vertical lines. (**B**) Line graphs of temporal phosphorylation levels of sites within each of the three clusters. Color scale represents the distance from the mean. The number of single phosphorylation sites is indicated. (**C**) Sequence logos for over and underrepresented

*Figure 4 continued on next page*

*Figure 4 continued*

motifs within the three clusters. Threonine with proline in the +1 position followed by basic amino acids is overrepresented in cluster 3, which is dephosphorylated after MI. In contrast, cluster 3 is depleted for serine followed by acidic amino acids. Instead, cluster 1, which is more stably phosphorylated, is enriched for serine but depleted for threonine as the phosphoacceptor. (D) Cumulative frequency distribution of phosphopeptides with proline-directed serine or threonine phosphorylation. Significant differences in the population distribution as determined by Kolmogorov-Smirnov (KS) statistics (****p<0.0001). Only cluster 2 shows a significant difference in the dephosphorylation of SP vs TP phosphorylation sites. (E) Cumulative frequency distribution of phosphopeptides with proline-directed threonine phosphorylation with different amino acids in the +2 position. Only the most significant differences are indicated (****p<0.0001). (F) The average phosphorylation abundance of sites that peak in MI, comparing threonine (black bar) vs serine (gray bars) phosphorylation sites across stages as determined by phosphoproteomics. (G) Quantification of western blots with an antibody recognizing phosphorylated TPxK vs K/HpSP, revealing distinct behaviors at the meiosis I/meiosis II (MI/MII) transition for these phosphorylation sites.

The online version of this article includes the following source data and figure supplement(s) for figure 4:

**Figure supplement 1.** Differential behavior of serine and threonine phosphorylation sites.
**Figure supplement 1—source data 1.** Differential behavior of serine and threonine phosphorylation sites.
**Figure supplement 2.** Analysis of pSPP and pTPP phosphorylation site motifs.
**Figure supplement 2—source data 1.** Analysis of pSPP and pTPP phosphorylation site motifs.

sites at the MI/MII transition, with threonine residues being preferentially dephosphorylated and SP sites remaining phosphorylated.

In addition to differences in the phosphorylated residue (T vs S), we also found that the identity of surrounding residues correlated with phosphorylation behavior. Specifically, we identified an enrichment for basic amino acids starting in the +2 position (e.g., lysine) and a depletion of acidic amino acids (e.g., aspartic and glutamic acid) (*Figure 4C*). For example, TP sites with a small non-polar (A or G) or basic (K or R) amino acid were significantly more dephosphorylated than TP with acidic amino acids (E or D) or proline (*Figure 4E*, *Figure 4—figure supplement 1A*). Importantly, we note that the sites preferentially dephosphorylated at the MI/MII transition (cluster 3, TP followed by basic or nonpolar amino acids) match the sequence preferences of the phosphatase PP2A-B55, which we and others have recently reported (*Cundell et al., 2016*; *Holder et al., 2020*; *Kruse et al., 2020*; *McCloy et al., 2015*; *Touati et al., 2019*). These observations raise the possibility that PP2A-B55 drives the striking difference in behavior between TP and SP sites at the MI/MII transition.

PP2A-B55 inhibition is required for meiotic resumption from prophase I into MI in response to hormonal stimulation (*Hara et al., 2012*; *Okumura et al., 2014*), but a role for PP2A-B55 at the MI/MII transition has not been defined directly. To determine the activation state of PP2A-B55 at the MI/MII transition, and whether it could selectively dephosphorylate TP residues, we assessed the conserved PP2A-B55 regulatory pathway in our MS datasets. Cdk1 phosphorylation activates Great-wall kinase, which then phosphorylates and activates the B55 inhibitor ARPP19 (*Figure 5A*; *Hara et al., 2012*; *Okumura et al., 2014*). Therefore, a drop in cyclin B levels and CDK activity would drive the reactivation of PP2A-B55. Indeed, following MI, our dataset reveals a decrease in cyclin B levels and a corresponding reduction in the activating phosphorylation on Great-wall kinase (T204) and ARPP19 (S106). This would result in the release of PP2A-B55 to dephosphory-late its substrates at the MI/MII transition (*Figure 5B*, *Figure 3—figure supplement 5C*). Together, these observations suggest that PP2A-B55 is reactivated at the MI/MII transition to preferentially dephosphorylate specific substrates with TP/basic consensus motifs, through a conserved pathway involving ARPP19, Greatwall, and CDK/cyclin B (*Figure 5B*).

## Selective dephosphorylation of TP vs SP residues is a conserved feature of meiosis

The selective dephosphorylation behavior that we observed suggests an important role for PP2A-B55 at the MI/MII transition. Preferential TP dephosphorylation has been previously reported to cre-ate a temporal order to the cellular events during mitotic exit (*Cundell et al., 2016*; *Hein et al., 2017*; *McCloy et al., 2015*; *Touati et al., 2019*). However, our work suggests that, in addition to temporal regulation of dephosphorylation at mitotic exit, preferential dephosphorylation of these motifs could provide a mechanism to specify the two meiotic divisions in oocytes. To test whether differential dephosphorylation is a conserved feature of meiosis, we reanalyzed a published phos-phoproteomics dataset of early development in *Xenopus laevis* oocytes (*Peuchen et al., 2017*).

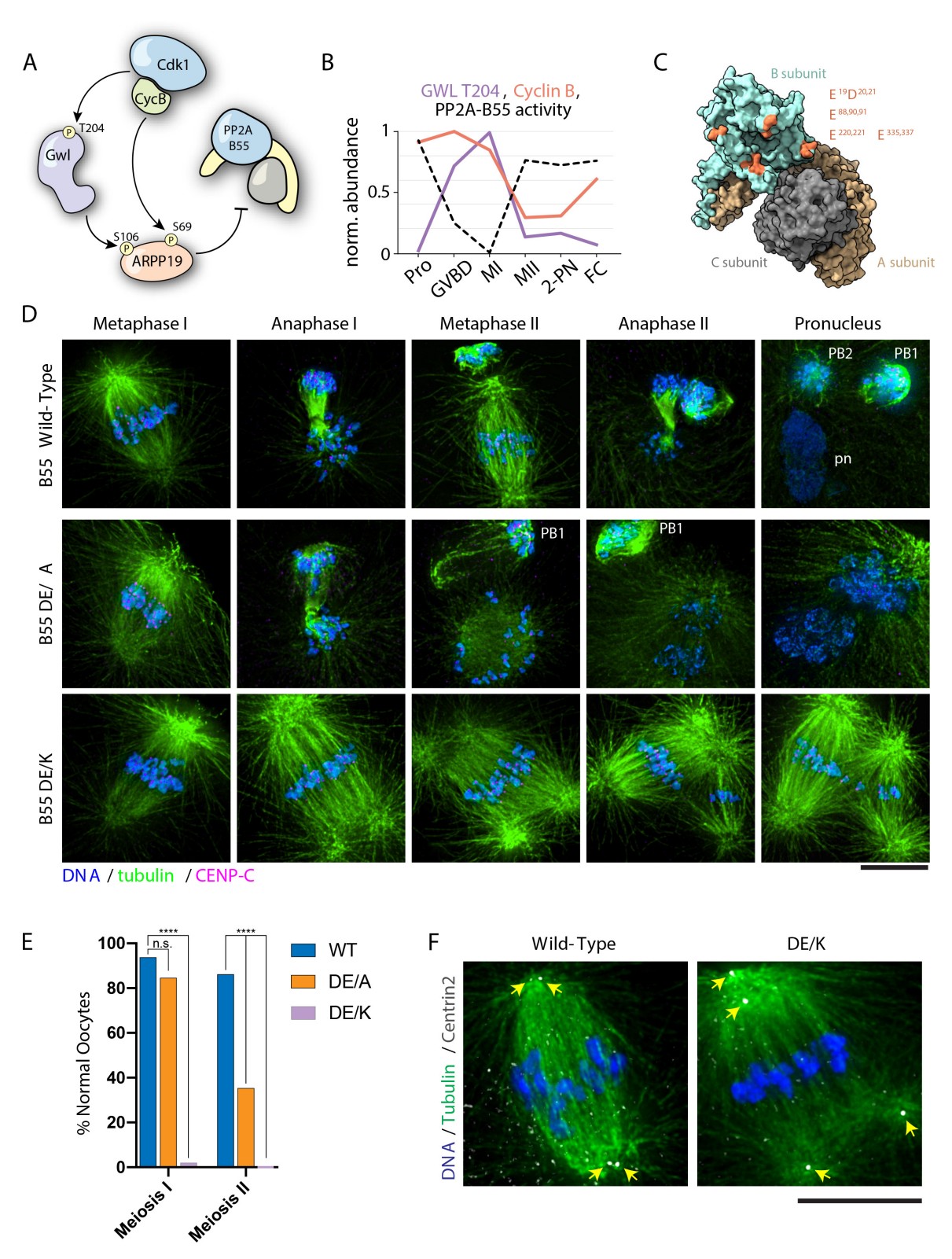

**Figure 5.** PP2A-B55 substrate specificity is required for the MI/MII transition. (**A**) Schematic of the Cdk-Gwl-ARPP19 pathway regulating PP2A-B55. (**B**) Relative phosphorylation levels of Gwl T204 and protein abundance of cyclin B. PP2A-B55 activity trace values are derived from the substrates in *Figure 5—figure supplement 3A–C* (1-mean phosphorylation levels). (**C**) Crystal structure of PP2A holoenzyme with the B subunit colored cyan. Mutated residues within the acidic surface are indicated in orange. (**D**) Immunofluorescence of meiotic time course of oocytes expressing wild-type or

*Figure 5 continued*

mutant B55 constructs. In contrast to wild-type oocytes, DE/A mutants successfully complete MI but fail in MII. The DE/K mutant oocytes successfully form the first meiotic spindle but fail to undergo homologous chromosome or sister centromere separation. The spindle poles do separate, ultimately resulting in two semi-distinct spindles. Microtubules were scaled nonlinearly. (E) Percentage of oocytes successfully completing MI and MII (control n = 65, DE/A n = 65, DE/K n = 49 oocytes; ****p<0.0001 by Fisher's exact test). (F) Centrin2 staining of B55 wild-type- and DE/K mutant-expressing oocytes stained for Centrin2 reveals centriole separation at the spindle poles. Scale bars = 10 μm.

The online version of this article includes the following figure supplement(s) for figure 5:

**Figure supplement 1.** Conservation of threonine-serine behaviors in *Xenopus laevis*.
**Figure supplement 2.** Mutant PP2A-B55 expression disrupts prophase I arrest.
**Figure supplement 3.** Phosphorylation behavior of conserved PP2A-B55 substrates.

Although the samples collected in this previous study did not have the temporal resolution to definitively capture the different stages of meiosis or the MI/MII transition, we used conserved inhibitory phosphorylation events on Cdk2 and PP1 to align these time points to corresponding samples from our analysis of synchronized sea star oocytes (*Figure 5—figure supplement 1A,B*). With this analysis, we identified time points that appear to correspond to an MI sample and an MII sample. Strikingly, we found that the phosphorylation sites that decrease in abundance between these samples in *Xenopus* oocytes were also enriched for TP-basic sequences but depleted for SP-acidic sequences (*Figure 5—figure supplement 1C*). Thus, based on this cross-species comparison, the differential behavior of SP and TP phosphorylation may represent a conserved regulatory mechanism for meiosis in animals and suggests that PP2A-B55 is an important conductor of the MI/MII transition.

## Selective dephosphorylation by PP2A-B55 is required for the MI/MII transition

Our data are consistent with a model in which PP2A-B55 reactivation at the MI/MII transition drives the selective dephosphorylation of substrates to reorganize the cell division machinery and distinguish these divisions. We therefore sought to test the functional contribution of PP2A-B55 to the MI/MII transition. As the small molecule used to inhibit PP2A also inactivates other phosphoprotein phosphatases (e.g., *Figure 3—figure supplement 7*), we instead altered its specificity through mutations designed to disrupt B55 interactions with the downstream basic patch in its substrates (*Cundell et al., 2016*; *Xu et al., 2008*; *Figure 5C*). We generated mutations in complementary acidic residues in B55, changing these to alanine or creating charge-swap substitutions of these sites to lysine. Compared to wild-type B55, ectopic expression of the alanine mutant (DE/A) induced spontaneous GVBD and apoptosis in prophase I-arrested oocytes (*Figure 5—figure supplement 2A,B*). This result, along with the effect that we observed following calyculin A treatment (*Figure 3—figure supplement 6A,C*), further supports our model in which PP2A is specifically required to maintain the prophase I arrest.

We next tested the functional contributions of B55 to meiosis. Following hormonal stimulation, the expression of either of the B55 mutants had a potent dominant effect on meiotic progression. Oocytes expressing the alanine mutant (DE/A) successfully assembled the MI spindle and completed cytokinesis of polar body I. However, events after MI did not occur, with the MII spindles failing to form properly and displaying a ball-like morphology. Furthermore, anaphase and cytokinesis did not progress normally, resulting in a reduced rate of polar body II extrusion (*Figure 5D,E*). Oocytes expressing the charge-swap mutations (DE/K) displayed a more severe phenotype: the first meiotic spindle formed normally, but remained arrested, with the bivalent chromosomes maintaining cohesion and the co-oriented centromeres remaining fused, even at time points where wild-type control oocytes completed MII. However, although normal anaphase I and cytokinesis did not occur, the spindle poles eventually separated, ultimately resulting in two semi-distinct spindles. Staining for Centrin2 revealed that pole fragmentation was due to the separation of the pair of centrioles in the spindle pole (*Figure 5F*). Therefore, alterations to PP2A-B55 substrate specificity allow MI events to occur, but prevent changes to the cell division machinery that occur at the MI/MII transition. We, therefore, propose that PP2A-B55 serves as an essential regulator of the MI/MII transition by selectively dephosphorylating substrates to achieve exit from MI, but retaining sites that must remain phosphorylated for MII to occur.

## Threonine-specific dephosphorylation is essential for the spatiotemporal control of PP2A-B55 substrates

The selective TP dephosphorylation at the MI/MII transition suggests a potential mechanism to encode MI- or MII-specific functions directly into substrates. The evolution and conservation of phosphorylation sites with higher or lower affinity for PP2A-B55 may provide temporal control for individual protein behaviors in meiosis. We identified several conserved phosphorylation sites on PRC1 (T470 in humans, T411 in sea star), TPX2 (T369 in humans, T508 in sea star), and Inner Centromere Protein (INCENP) (T59 in humans, T61 in sea star) (*Figure 5—figure supplement 3A–C*) that are known substrates of PP2A-B55 (*Cundell et al., 2016*; *Hein et al., 2017*; *Hümmer and Mayer, 2009*; *Jiang et al., 1998*). Consistent with the reactivation of PP2A-B55, these substrates display a stark decrease in phosphorylation following MI. In contrast to the behavior of these conserved PP2A-B55 substrate sites, we identified an SP phosphorylation event on Prc1 (S571), whose levels remain relatively constant during the meiotic divisions (*Figure 5—figure supplement 3D*). Furthermore, relative to other TP sites, an SP phosphorylation in Tpx2 (S625) was dephosphorylated to a lesser degree following MI (*Figure 5—figure supplement 3E*). To test the contributions of these sites, we focused on the chromosome passenger complex (CPC) subunit INCENP, which contains both stable serine phosphorylations as well as MI-specific threonine phosphorylations (*Figure 6A*). During anaphase of mitosis, the CPC transitions from the inner centromere to the central spindle, where it is required for cytokinesis (*Carmena et al., 2012*; *Kaitna et al., 2000*). However, the localization dynamics and impact of phosphorylation are not well defined in meiosis. Using a green fluorescent protein (GFP) fusion construct, we found that INCENP-GFP localized to centromeres in metaphase of MI, but then translocated to the central spindle at anaphase of MI. In MII, INCENP localization to centromeres was reduced, but it then returned at the midzone in anaphase II. As the maternal pronucleus formed at meiotic exit, INCENP localized to nucleolar structures (*Figure 5—figure supplement 3F*).

To test whether PP2A-B55 activity is required for the localization dynamics of INCENP, we co-expressed INCENP-GFP with either wild-type B55 or the B55 DE/A mutant. In wild-type B55-expressing oocytes, INCENP correctly translocated to the central spindle in anaphase of MI. In contrast, expression of the B55 DE/A mutant significantly reduced translocation of INCENP to the central spindle (*Figure 6B,C*). Therefore, the selective activity of PP2A-B55 is required for the proper localization behavior of INCENP at the MI/MII transition. Consistent with this model, we found that phosphorylation on INCENP T61 sharply decreased following MI, whereas other proline-directed sites on INCENP remained phosphorylated throughout the oocyte-to-embryo transition (*Figure 6A*, *Figure 5—figure supplement 3C*). Sequence alignment of sea star T61 with human INCENP T59 revealed the presence of a conserved downstream basic patch, supporting its potentially conserved dephosphorylation by PP2A-B55 (*Figure 5—figure supplement 3C*). In contrast, T506 remains relatively stable, likely due to the presence of downstream acidic amino acids, thereby disfavoring PP2A-B55 association (*Figure 6A*).

Finally, to test the role of INCENP T61 residue in MI, we used GFP fusion constructs with either wild-type PmINCENP or T61 replaced with serine (T61S), which is typically considered to be a conservative change that would preserve protein function. Wild-type INCENP localized to centromeres in metaphase of MI, but then relocalized to the central spindle at anaphase of MI. In contrast, INCENP(T61S)-GFP failed to translocate to the central spindle in anaphase of MI and remained at high levels at centromeres (*Figure 6D,E*), consistent with prior work in mitosis (*Goto et al., 2006*; *Hümmer and Mayer, 2009*). The retention of INCENP at centromeres in anaphase I suggests that this residue remains phosphorylated at the MI/MII transition when substituted with serine, a lower affinity substrate for PP2A-B55. Thus, the usage of serine vs threonine, modulated by adjacent charged amino acids, can directly encode a differential response to a common set of kinases and phosphatases, enabling rewiring events at key cell cycle transitions. Through this paradigm, the behaviors of individual proteins may be temporally coordinated with meiotic cell cycle progression to achieve specific behaviors.

## Discussion

In this work, we define an extensive program of phosphorylation changes during the oocyte-to-embryo transition, spanning the complete developmental window from a prophase I arrest to the

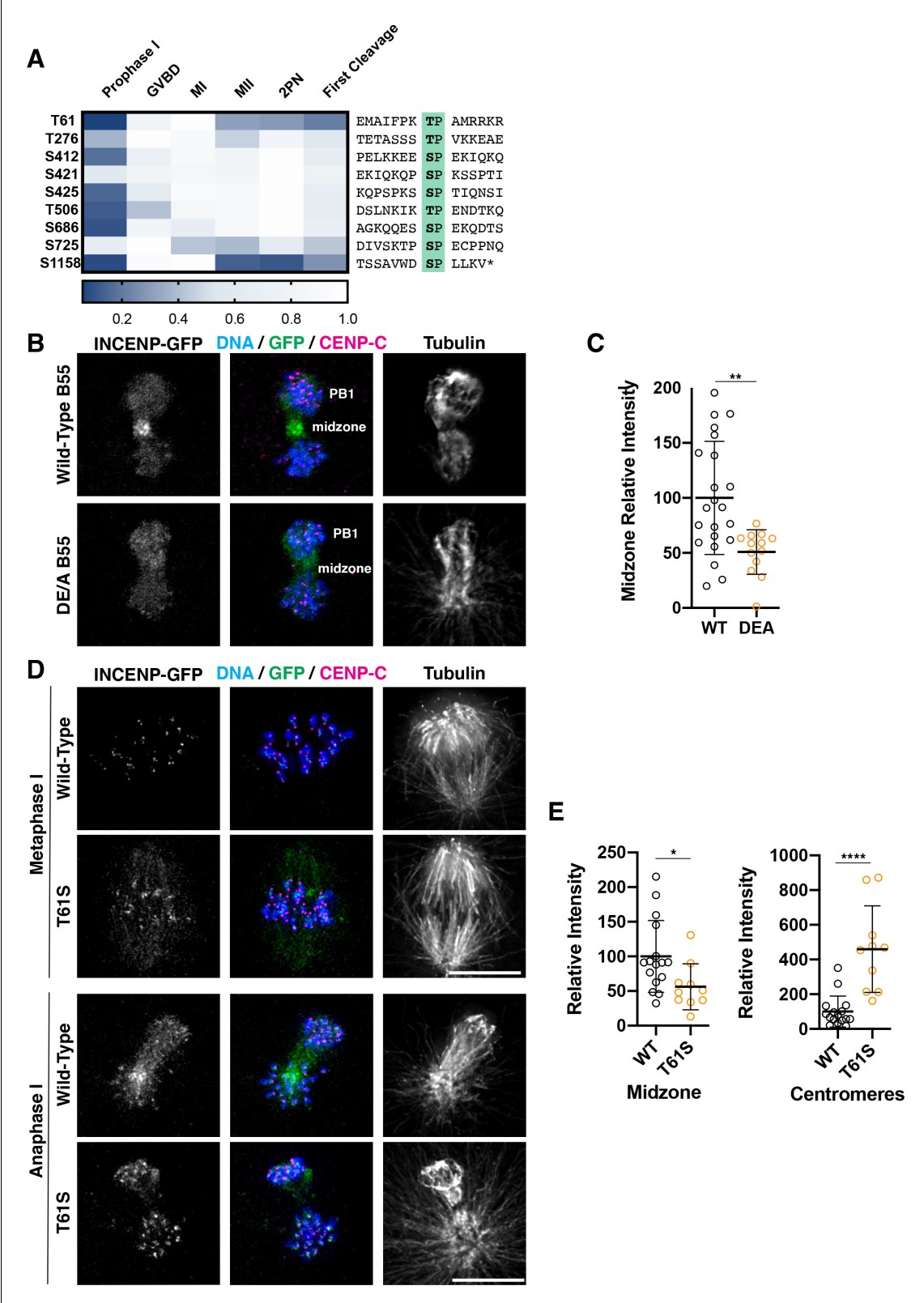

**Figure 6.** Dephosphorylation of a conserved threonine is necessary for INCENP localization. (A) Heatmap representation of relative phosphorylation levels of sites identified by phosphoproteomics. Phospho-centered amino acid sequences are provided on the right. (B) Immunofluorescence images of oocytes expressing INCENP-GFP in meiosis I along with either wild-type B55 or the DE/A mutant. DE/A expression reduces translocation of INCENP-GFP to the central spindle. Images are scaled individually to aid in visualization of localization changes. (C) Pixel intensity quantification of the unscaled

*Figure 6 continued on next page*

*Figure 6 continued*

midzone INCENP signal with wild-type or DE/A B55 expression (control n = 22, DE/A n = 13 oocytes; error bars represent mean and standard deviation; **p = 0.0014, Mann-Whitney test). (D) Immunofluorescence images of oocytes expressing wild-type INCENP-GFP or INCENP^T61S. Wild-type INCENP translocates from centromeres to the central spindle in anaphase, whereas INCENP^T61S increases at centromere localization but not at the central spindle. Images are scaled individually to aid in visualization of localization changes. (E) Pixel intensity quantitation of unscaled oocyte images at centromeres or central spindles in anaphase of meiosis I (WT n = 16, T61S n = 10 oocytes; error bars represent mean and standard deviation; *p = 0.0122, ****p<0.0001, Mann-Whitney test).

first embryonic cleavage. Using TMT-based proteomics, phosphoproteomics, and functional approaches, we found that overall protein levels are stable but that selected cell cycle regulators, including cyclins A and B, must undergo new protein synthesis for progression through meiosis. This general stability of the proteome across the oocyte-to-embryo transition in sea stars is consistent with prior results in oocytes of the vertebrate *X. laevis* (*Peuchen et al., 2017*; *Presler et al., 2017*). Therefore, post-translational modification instead serves as the primary and conserved control mechanism for meiosis and the transition to early development. Indeed, our analysis has uncovered a complex landscape of phosphorylation and dephosphorylation that underlies this developmental period. We found that the prophase I arrest is characterized by low overall phosphorylation and that maintaining this arrest requires phosphatase activity. Strikingly, although serine and threonine residues are often considered as conserved and interchangeable, we observed distinct behaviors for TP and SP phosphorylation at the MI/MII transition, with threonine sites being preferentially dephosphorylated. Such differential dephosphorylation suggests a novel paradigm for the regulatory control of oocytes, which must rapidly transition between the two meiotic divisions for the specialized meiotic cell cycle.

Our analysis reveals that, upon the decrease in Cdk activity at the MI/MII transition, selective dephosphorylation of TP vs SP sites by PP2A-B55 is essential for meiotic progression. By analzying the TP sites that are dephosphorylated after MI, we identified an enrichment for basic amino acids starting in the +2 position, matching the known consensus for PP2A-B55 substrates (*Cundell et al., 2016*; *Kruse et al., 2020*). Moreover, we found that PP2A-B55 is re-activated at the MI/MII transition, based on phosphorylation signatures of its inhibitory pathway, including Greatwall and ARPP19, as well as conserved PP2A-B55 substrates. Our results are consistent with prior studies in a related sea star reporting complete cyclin B degradation followed by rapid resynthesis to a lower level for MII and pronuclear fusion, resulting in the dephosphorylation of Greatwall kinase following MI and during first cleavage (*Hara et al., 2012*; *Okano-Uchida et al., 1998*; *Tachibana et al., 2008*). Thus, the temporal profile of PP2A-B55 activity leaves it poised to play an essential role in the MI/MII transition for specializing the meiotic cell cycle and division machinery.

Our work supports the emerging picture that threonine and serine phosphorylation is not interchangeable (*Cundell et al., 2016*; *Deana et al., 1982*; *Deana and Pinna, 1988*; *Hein et al., 2017*; *Pinna et al., 1976*) but instead represents an important regulatory mechanism to temporally control cellular processes. In cultured mitotic cells, threonine dephosphorylation is important for timely mitotic exit. For example, inhibitory phosphorylations on the APC/C regulator Cdc20 are conserved as threonines, whereas serine substitution mutants display delayed dephosphorylation and delayed Cdc20 activation (*Hein et al., 2017*). Our work extends this model into a new physiological context, wherein differential dephosphorylation provides not just a kinetic delay but also confers distinct behaviors for the two different meiotic divisions. For example, in our mutant analysis, substitution of threonine 61 for serine on INCENP, which is subject to inhibitory phosphorylation by Cdk1, resulted in its failure to translocate to the central spindle in anaphase of MI. This phosphorylation code is further modulated by the charge of adjacent amino acids, with positively charged residues favoring PP2A-B55 association. These findings reveal the importance for temporally coordinated dephosphorylation during meiotic progression, such that critical events including chromosome segregation and cytokinesis are properly synchronized with other cellular events. Such differential dephosphorylation represents a novel paradigm for the regulatory control of oocytes, which must rapidly transition between the two meiotic divisions for the specialized meiotic cell cycle.

These results point to PP2A-B55 as a critical conductor of cell cycle progression in oocytes. In diverse species, ARPP19/ENSA is required for exit from prophase I arrest in response to hormonal stimulation by inhibiting PP2A-B55 (*Dupré et al., 2013*; *Dupré et al., 2014*; *Matthews and Evans,*

*2014*; *Okumura et al., 2014*; *Von Stetina et al., 2008*). Intriguingly, although there is considerable conservation among the downstream phosphorylation steps associated with Cdk1 activation and meiotic progression, the upstream triggers of meiotic cell cycle resumption have diverged across evolution. For example, numerous species, including mammals, sea stars, and cnidarians (jellyfish), all utilize GPCRs that respond to hormonal signals and undergo communication with surrounding somatic ovarian cells to propagate signals (*Jaffe and Norris, 2010*; *Quiroga Artigas et al., 2020*; *Tadenuma et al., 1992*). However, the hormone/receptor pairs differ amongst species. In mice, luteinizing hormone acts upon receptors expressed in somatic cells and induces them to reduce the production of cyclic guanosine monophosphate (cGMP) and close gap-junction connections with the oocytes. This leads to a decrease in cyclic adenosine monophosphate (cAMP) levels in the oocyte, a drop in protein kinase A (PKA) activity, and the subsequent release from meiotic arrest by activation of Cdc25 (*Conti and Chang, 2016*; *Norris et al., 2009*). Similarly, in *X. laevis,* a drop in PKA activity is associated with meiotic resumption downstream of progesterone stimulation, although the nature of the roles of PKA and cAMP is disputed (*Dupré et al., 2014*; *Nader et al., 2016*). In echinoderms, meiotic resumption is stimulated by the hormone 1-MeAd, which is synthesized by surrounding somatic cells and acts upon an unidentified GPCR, leading to the activation of the trigger kinase SGK (*Kishimoto, 2018*). Although these upstream steps are divergent, they still result in the downstream activation of Cdc25 and inhibition of Myt1, as in vertebrate oocyte systems.

Our data from sea stars indicate that phosphatase activity is critical not only for maintaining the prophase I arrest but also to drive the transition between the two meiotic divisions. In support of a potentially conserved role for PP2A-B55 in the MI/MII transition, prior work in mouse oocytes found that the Greatwall ortholog Mastl is essential to proceed past MI (*Adhikari et al., 2014*). CDK and MAPK, combined with a wave of opposing PP2A-B55 activity at the MI/MII transition, provide a pacemaker for meiotic cell cycle progression. By encoding PP2A-B55 affinity into substrates, different temporal phosphorylation profiles can be achieved yielding subsets of cell division proteins that are constitutively active but others that are uniquely modified in MI or MII. The rapidity of the two meiotic divisions (separated by only 30 min in sea stars) places a selective pressure for specific substrates to be promptly dephosphorylated. In contrast, other sites may need to remain phosphorylated to prevent exit into interphase, creating evolutionary pressure for conservation as serine. The ability to segregate heritable material through meiosis is essential for organismal fitness. We propose that evolution has fine-tuned substrate dephosphorylation by selecting for amino acid sequences that favor or disfavor the interaction with PP2A-B55, thereby enabling precise temporal coordination of events in the oocyte-to-embryo transition.

# Materials and methods

**Key resources table**

| Reagent type (species) or resource | Designation | Source or reference | Identifiers | Additional information |
|---|---|---|---|---|
| Strain, strain background (*Patiria miniata*) | | South Coast Bio, LLC | | Wild-caught animals |
| Antibody | Anti-CENP-C (rabbit polyclonal) | *Swartz et al., 2019* | | IF(1:5000) |
| Antibody | Anti-cyclin B (rabbit polyclonal) | *Ookata et al., 1992* | | WB(1:1000) |
| Antibody | Anti-alpha tubulin, DM1alpha (mouse monoclonal) | Sigma-Aldrich | Cat# T9026, RRID:AB_477593 | IF(1:5000) |
| Antibody | Anti-phoshpo-Cdk1 Y15 | Cell Signaling Technologies | Cat# 4539 | WB(1:3000) |
| Antibody | Anti-phospho ERK1/2 T202/Y204 | Cell Signaling Technologies | Cat# 9101 | WB(1:3000) |
| Antibody | Anti-pTPxK CDK consensus | Cell Signaling Technologies | Cat# 14371 | WB(1:3000) |

*Continued on next page*

*Continued*

| Reagent type (species) or resource | Designation | Source or reference | Identifiers | Additional information |
|---|---|---|---|---|
| Antibody | Anti-(K/H)pSP CDK consensus | Cell Signaling Technologies | Cat# 9477 | WB(1:3000) |
| Antibody | Anti-pTPP CDK consensus | Cell Signaling Technologies | Cat# 5757 | WB(1:3000) |
| Antibody | GFP booster | Chromotek | gba488-100, RRID:AB_2631386 | IF(1:500) |
| Recombinant DNA reagent | pZS281 (plasmid) | This paper | To be deposited to Addgene | Wild-type B55-GFP (pCS2+eight backbone) |
| Recombinant DNA reagent | pZS282 (plasmid) | This paper | To be deposited to Addgene | DE/A mutant B55-GFP (pCS2+eight backbone) |
| Recombinant DNA reagent | pZS283 (plasmid) | This paper | To be deposited to Addgene | DE/K mutant B55-GFP (pCS2+eight backbone) |
| Recombinant DNA reagent | pZS294 (plasmid) | This paper | To be deposited to Addgene | B55-mCherry (pCS2+eight backbone) |
| Recombinant DNA reagent | pZS295 (plasmid) | This paper | To be deposited to Addgene | DE/A mutant B55-mCherry (pCS2+eight backbone) |
| Recombinant DNA reagent | pZS13 (plasmid) | This paper | To be deposited to Addgene | INCENP-GFP (pCS2+eight backbone) |
| Recombinant DNA reagent | pZS259 (plasmid) | This paper | To be deposited to Addgene | INCENP-GFP T61S mutant (pCS2+eight backbone) |
| Recombinant DNA reagent | GST-ARPP19 | This paper | To be deposited to Addgene | For protein expression |
| Recombinant DNA reagent | pCS2+eight backbone | *Gökirmak et al., 2012* | RRID:Addgene_34952 | Pol III-based shRNA backbone |
| Sequence-based reagent | Standard Control Morpholino | Gene-Tools, LLC | | CCT CTT ACC TCA GTT ACA ATT TAT |
| Sequence-based reagent | Cyclin A morpholino | Gene-Tools, LLC | Morpholino antisense oligo | TTCACTTTGTT CCCGAGATTAAC |
| Sequence-based reagent | Cyclin B morpholino | Gene-Tools, LLC | Morpholino antisense oligo | TAACCAATGCGA GTTCCGAGGAG |
| Commercial assay or kit | mMessage mMachine SP6 in vitro transcription kit | Invitrogen | Cat# AM1340 | |
| Commercial assay or kit | Poly(A) tailing kit | Invitrogen | Cat# AM1350 | |
| Commercial assay or kit | Can Get Signal Immunoreaction Enhancer Solution | Cosmo Bio USA | Cat# TYB-NKB-101T | |
| Chemical compound, drug | 1-Methyladenine | Acros Organics | AC20131-1000 | |
| Chemical compound, drug | Calyculin A | Santa Cruz Biotechnology | SC-24000A | PP1 and PP2A inhibitor |
| Chemical compound, drug | Emetine | Sigma Aldrich | E2375-500MG | Translation inhibitor |
| Chemical compound, drug | Tautomycetin | Gift from Dr. Richard Honkanen | | PP1 inhibitor |
| Software, algorithm | SPSS | SPSS | RRID:SCR_002865 | |
| Other | Hoechst 33342 stain | Life Technologies | Cat# H1399 | (2 μg/ml) |

*Continued on next page*

*Continued*

| Reagent type (species) or resource | Designation | Source or reference | Identifiers | Additional information |
|---|---|---|---|---|
| Other | Prolong Gold Antifade Reagent | Life Technologies | P36930 | |
| Commercial assay or kit | TMT10plex Isobaric Label Reagent Set plus TMT11-131C Label Reagent | Thermo Scientific | A34808 | |
| Commercial assay or kit | High-Select Fe-NTA Phosphopeptide Enrichment Kit | Thermo Scientific | A32992 | |

## Experimental models and subject details

Sea stars (*P. miniata*) were wild-caught by South Coast Bio Marine (http://scbiomarine.com/) and kept in artificial seawater aquariums at 15°C. Intact ovary and testis fragments were surgically extracted as previously described (*Swartz et al., 2019*). Oocyte samples for proteomics were collected from a single female within the same season to maximize synchrony between time courses. Stereotypical meiotic timings were visually confirmed in live oocytes using GVBD and polar body emission as metrics, before snap freezing samples in liquid nitrogen. The synchrony data in Figure S1B were performed with oocytes from a second animal collected at the same season and location that was confirmed to undergo meiosis at a similar rate. 293T cells were obtained from commercial sources and were routinely tested for mycoplasma by polymerase chain reaction (PCR).

## Ovary and oocyte culture

Ovary fragments were maintained in artificial seawater containing 100 units/ml pen/strep solution. Intact ovary fragments were cultured this way for up to 1 week until oocytes were needed, with media changes every 2–3 days. Isolated oocytes were cultured for a maximum of 24 hr in artificial seawater with pen/strep. To induce meiotic re-entry, 1-MeAd (Acros Organics) was added to the culture at a final concentration of 10 µM. For fertilization, extracted sperm was added to the culture at 1:1,000,000 dilution prior to emission of the first polar body. For emetine treatments (*Figure 2*), oocytes were pre-treated with 10 µM emetine (Sigma-Aldrich) for 30 min prior to hormonal stimulation.

## Mass-spectrometry sample preparation

Oocytes were collected by centrifugation at 200 x*g* and resuspending pellets and washed one time with wash buffer (50 mM 4-(2-hydroxyethyl) -1-piperazineethanesulfonic acid (HEPES), pH 7.4, 1 mM ethylene glycol tetraacetic acid (EGTA), 1 mM MgCl$_2$, 100 mM KCl, 10% glycerol), pelleted with all excess buffer removed, and snap frozen in liquid nitrogen. Samples were lysed and proteins were digested into peptides with trypsin. Oocyte pellets were lysed in ice-cold lysis buffer (8 M urea, 25 mM Tris-HCl, pH 8.6, 150 mM NaCl, phosphatase inhibitors (2.5 mM beta-glycerophosphate, 1 mM sodium fluoride, 1 mM sodium orthovanadate, 1 mM sodium molybdate), and protease inhibitors (one mini-Complete EDTA-free tablet per 10 ml lysis buffer; Roche Life Sciences)) and sonicated three times for 15 s each with intermittent cooling on ice. Lysates were centrifuged at 15,000 x*g* for 30 min at 4°C. Supernatants were transferred to a new tube and the protein concentration was determined using bicinchoninic acid (BCA) assay (Pierce/ThermoFisher Scientific). For reduction, dithiothreitol (DTT) was added to the lysates to a final concentration of 5 mM and incubated for 30 min at 55°C. Afterwards, lysates were cooled to room temperate and alkylated with 15 mM iodoacetamide at room temperature for 45 min. The alkylation was then quenched by the addition of an additional 5 mM DTT. After sixfold dilution with 25 mM Tris-HCl, pH 8, the samples were digested overnight at 37°C with 1:100 (w/w) trypsin (Promega). The next day, the digest was stopped by the addition of 0.25% trifluoroacetic acid (TFA) (final v/v), centrifuged at 3500 x*g* for 15 min at room temperature to pellet precipitated lipids, and peptides were desalted (*Zecha et al., 2019*). Peptides were lyophilized and stored at −80°C until further use.

TMT labeling was carried out as previously described (*Zecha et al., 2019*). For proteomics analysis, an aliquot containing ~ 100 µg of peptides was resuspended in 133 mM HEPES (SIGMA), pH 8.5, TMT reagent (Thermo Scientific) stored in dry acetonitrile (ACN) (Burdick and Jackson) was added, and vortexed to mix reagents and peptides. After 1 hr at room temperature, an aliquot was withdrawn to check for labeling efficiency by LC-MS³ analysis, while the remaining reaction was stored at −80°C. Once labeling efficiency was confirmed to be at least 95%, each reaction was quenched by addition of ammonium bicarbonate to a final concentration of 50 mM for 10 min, mixed, diluted with 0.1% TFA in water, and desalted. The desalted multiplex was dried by vacuum centrifugation and separated by offline pentafluorophenyl (PFP)-based reverse-phase high-performance liquid chromatography (HPLC) fractionation as previously described (*Grassetti et al., 2017*).

For phosphoproteomic analysis, ~4 mg of peptides was enriched for phosphopeptides using a nitrilotriacetic acid (Fe-NTA) phosphopeptide enrichment kit (Thermo Scientific) according to instructions provided by the manufacturer and desalted. Phosphopeptides were labeled with TMT reagent and labeling efficiency was determined as described for the protein sample. Once labeling efficiency was confirmed to be at least 95%, each reaction was quenched with 5 mM ammonium bicarbonate, mixed, diluted with 0.1% trifluoroacetic acid (TFA) in water, and desalted. The desalted multiplex was dried by vacuum centrifugation and separated by offline PFP-based reverse-phase HPLC fractionation as previously described (*Grassetti et al., 2017*).

Proteomic and phosphoproteomic analyses of oocyte-to-embryo time courses were performed on an Orbitrap Fusion mass spectrometer (Thermo Scientific) equipped with an Easy-nLC 1000 (Thermo Scientific) (*Senko et al., 2013*). Peptides were resuspended in 8% methanol/1% formic acid and separated on a reverse-phase column (40 cm length, 100 µm inner diameter, ReproSil, C18 AQ 1.8 µm 120 Å pore) pulled and packed in-house across a 2-hr gradient from 3% acetonitrile/0.0625% formic acid to 37% acetonitrile/0.0625% formic acid. The Orbitrap Fusion was operated in data-dependent, SPS-MS3 quantification mode (*McAlister et al., 2014*; *Ting et al., 2011*). For proteomics analysis, an Orbitrap MS1 scan was taken (scan range = 350–1300 m/z, R = 120K, automatic gain control (AGC) target = 2.5e5, max ion injection time = 100 ms). This was followed by a data-dependent iontrap MS2 scan on the most abundant precursors for 3 s in rapid scan modem (AGC target = 1e4, max ion injection time = 40 ms, collision-induced dissociation (CID) collision energy = 32%), followed by an Orbitrap MS3 scan for quantification (R = 50K, AGC target = 5e4, max ion injection time = 100 ms, higher energy collision dissociation (HCD) collision energy = 60%, scan range = 110–750 m/z, synchronous precursors selected = 10). For phosphoproteomics analysis, an Orbitrap MS1 scan was taken (scan range = 350–1300 m/z, R = 120K, AGC target = 3.5e5, max ion injection time = 100 ms), followed by data-dependent Orbitrap MS2 scans on the most abundant precursors for 3 s. Ion selection (to ensure nominally informative peptide length for unambiguous sequence assignment); charge state = 2: minimum intensity 2e5, precursor selection range 625–1200 m/z; charge state 3: minimum intensity 3e5, precursor selection range 525–1200 m/z; charge states 4 and 5: minimum intensity 5e5 (quadrupole isolation = 0.7 m/z, R = 30K, AGC target = 5e4, max ion injection time = 80 ms, CID collision energy = 32%). This was followed by an Orbitrap MS3 scan for quantification (R = 50K, AGC target = 5e4, max ion injection time = 100 ms, HCD collision energy = 65%, scan range = 110–750 m/z, synchronous precursors selected = 5).

The phosphoproteomic analysis of calyculin A-treated occytes was performed on the Orbitrap Lumos mass spectrometer, wherein an Orbitrap MS1 scan was taken (scan range = 350–1250 m/z, R = 120K, AGC target = 2.5e5, max ion injection time = 50 ms), followed by data-dependent Orbitrap MS2 scans on the most abundant precursors for 2 s. Ion selection (to ensure nominally informative peptide length for unambiguous sequence assignment); charge state = 2: minimum intensity 2e5, precursor selection range 650–1250 m/z; charge state 3: minimum intensity 3e5, precursor selection range 525–1250 m/z; charge states 4 and 5: minimum intensity 5e5 (quadrupole isolation = 1 m/z, R = 30K, AGC target = 5e4, max ion injection time = 55 ms, CID collision energy = 35%). This was followed by Orbitrap MS3 scans for quantification (R = 50K, AGC target = 5e4, max ion injection time = 100 ms, HCD collision energy = 65%, scan range = 100–500 m/z, synchronous precursors selected = 5).

High-resolution tandem mass spectra were searched using COMET with a precursor ion tolerance of +/-1 Dalton and a fragment ion tolerance of +/-0.02 m/z, static mass of 229.162932 on peptide N-termini and lysines and 57.02146 Da on cysteines, and a variable mass of 15.99491 Da on methionines against a target-decoy version of the *P. miniata* proteome sequence database. For

phosphoproteomics analysis, 79.96633 Da on serines, threonines, and tyrosines was included as an additional variable mass. The resulting peptide spectral matches (PSMs) were filtered to a <1% false discovery rate (FDR) using the target-decoy strategy with a typical precursor ion mass filter of +/-1.5 parts-per-million (PPM) mass accuracy and corresponding XCorr and dCn values. Quantification of spectra of liquid chromatography with tandem mass spectrometry (LC-MS/MS) was performed using an in-house-developed software. Probability of phosphorylation site localization was determined by PhosphoRS (*Taus et al., 2011*).

The *P. miniata* proteome sequence database was generated using published paired-end RNA-seq data for a *P. miniata* ovary sample (SRX445851/SRR1138705), with the Agalma transcriptome pipeline (*Dunn et al., 2013*). Each protein sequence from this transcriptome was assessed for similarity with known proteins in the NCBI nr (nonredundant) sequence database using NCBI BLAST. Transcriptome sequences were annotated with the accession and sequence similarity metrics of their top BLAST hits. This annotation procedure was repeated for an unannotated protein database sourced from EchinoBase to generate a composite database. Duplicate sequences and subsequences within the composite database were removed to reduce redundancy.

## Construct generation

*P. miniata* INCENP was identified using the genomic resources at echinobase.org and previously published ovary transcriptomes (*Kudtarkar and Cameron, 2017*; *Reich et al., 2015*). Wild-type INCENP was amplified from first-strand complementary DNA (cDNA) reverse transcribed from total ovary mRNA. T61S mutants were generated by overlap extension PCR. These cDNAs were then cloned into pCS2+eight as c-terminal GFP fusions (*Gökirmak et al., 2012*). Wild-type and mutant versions of B55 were synthesized (Twist Biosciences) and cloned into pCS2+eight with standard restriction enzyme methods.

## Oocyte microinjection

For expression of constructs in oocytes, plasmids were linearized with NotI to yield the linear template DNA. mRNA was transcribed in vitro using mMessage mMachine SP6 and the polyadenylation kit (Life Technologies), then precipitated using lithium chloride solution. Prophase I-arrested oocytes were injected horizontally in Kiehart chambers with approximately 10–20 picoliters of mRNA solution in nuclease-free water. B55 constructs were injected at 500 ng/µl, and INCENP constructs were injected at 1000 ng/µl. After microinjection, oocytes were cultured for 18–24 hr to allow time for the constructs to translate before 1-MeAd stimulation. Custom-synthesized cyclin morpholinos, or the Gene Tools standard control, were injected at 500 µM immediately before 1-MeAd stimulation (Gene Tools).

## Immunofluorescence, imaging, and immunoblots

Oocytes were fixed at various stages in a microtubule stabilization buffer as described previously (2% paraformaldehyde, 0.1% triton X-100, 100 mM HEPES, pH 7.0, 50 mM EGTA, 10 mM MgSO$_4$, 400 mM dextrose) for 24 hr at 4° C (*von Dassow et al., 2009*). The oocytes were then washed three times in phosphate-buffered saline (PBS) with 0.1% triton X-100 and blocked for 15 min in AbDil (3% bovine serum albumin (BSA), 1 X tris-buffered saline (TBS), 0.1% triton X-100, 0.1% sodium azide). Primary antibodies diluted in AbDil were then applied and the oocytes were incubated overnight at 4°C. Anti-CENP-C and alpha-tubulin (DM1α, Sigma) antibodies were used at 1:5000 ratio. DNA was stained with Hoechst. GFP booster (Chromotek) was used at 1:500 to improve the signal from GFP fusion constructs. The oocytes were compressed under coverslips in ProLong Gold antifade Mountant (Thermo Fisher). Images were collected with a DeltaVision Core microscope (Applied Precision/GE Healthsciences) with a CoolSnap HQ2 CCD camera and a x100 1.40 NA Olympus U-PlanApo objective. Confocal images (*Figure 2G*) were collected with a Yokogawa W1 spinning disk microscope. Images were processed with Fiji and scaled equivalently across conditions unless otherwise specified (*Schindelin et al., 2012*). For western blot analyses, oocytes were lysed in Laemmli samples buffer, separated by sodium dodecyl sulfate-polyacrylamide gel electrophoresis (SDS-PAGE), and analyzed by western blotting. Antibodies were diluted in 3% milk in TBST (TBS with Tween 20) and phosphatase inhibitors anti-phospho Cdk1 Y15 (Cell Signaling, #4539), anti-phospho ERK1/2 T202/Y204 (Cell Signaling, #9101; this antibody detects singly and dually phosphorylated ERK1/2 and the

total intensity of singly and dually phosphorylated ERK1/2 in sea stars suggests that Y193 is the dominant phosphoform), phospho CDK substrate [pTPXK] (Cell Signaling, #14371), [(K/H)pSP] (Cell Signaling, #9477), [pSPP] (Cell Signaling, #14390), and [pTPP] (Cell Signaling, #5757). Antibodies raised against phosphorylation sites on human proteins were evaluated for cross-reactivity and judged to do if (1) they only recognized one band, (2) the size of the band migrated at the expected molecular weight, and (3) the phosphorylation pattern matched the MS quantification. Anti-*P. pectinifera* cyclin B antibody was a gift from the Kishimoto laboratory (*Ookata et al., 1992*). Cyclin B western blotting was conducted by diluting the primary and secondary antibodies into Can Get Signal Immunoreaction Enhancer Solution as described by the manufacturer's instructions (Cosmo Bio USA).

## Protein expression

*P. miniata* ARPP19 was identified using the genomic resources at echinobase.org and previously published ovary transcriptomes (*Kudtarkar and Cameron, 2017*; *Reich et al., 2015*). Wild-type ARPP19 was amplified from first-strand cDNA reverse transcribed from total ovary mRNA and cloned into pCS2+8. ARPP19 was sub-cloned into pGEX4T3 and transformed into Rosetta (DE3). Expression, purification, and thiophosphorylation were performed as previously described (*Gharbi-Ayachi et al., 2010*).

## In vitro phosphatase assay

PP1c was diluted in assay buffer (0.15 M NaCl, 30 mM HEPES, pH 7.0, 1 mM DTT, 0.1 mg/ml BSA, 1 mM ascorbate, 1 mM $MnCl_2$) with a final assay concentration of 0.5 nM PP1c and 75 µM 6,8-difluoro-4-methylumbelliferyl phosphate (DiFMUP) (Invitrogen, D6567). 6,8-difluro-4-methylumbelliferone (DiFMU) was measured every minute for 45 min (Ex 360 nm, Em 460 nm) in the presence of indicated concentrations of calyculin A and tautomycetin.

## Quantification and statistical analysis

For the time-course proteomic analysis, peptide TMT intensities were summed on a per protein basis. Replicate multiplexes were normalized to each other based on an internal standard consisting of a mixture of oocytes from all investigated stages. Protein intensities were adjusted based on the total TMT reporter ion intensity in each channel. Individual protein intensity across each time course was scaled between 0 and 1. For the time-course phosphoproteomic analysis, replicate multiplexes were normalized to each other based on the internal standard. Phosphopeptide intensities were adjusted based on the total amount of protein input in the respective channel to account for differences in input and the median total TMT reporter ion intensity of the respective condition. Individual phosphopeptide intensity across each time course was scaled between 0 and 1. A Pearson correlation coefficient for the time-course data was calculated as pairwise correlation for proteins and phosphopeptides identified in two time-course series and as multiple correlations for proteins identified in three time-course series in Excel. A p-value was calculated from the Pearson correlation coefficient. Proteins quantified in all three time-course series were analyzed by hierarchical clustering using Euclidean distance and average linkage in Perseus. Phosphorylation sites with a localization score of 0.9 or higher and quantified with a correlation coefficient of 0.8 more (corresponding to a p-value of 0.05 or less) were analyzed by hierarchical clustering using Euclidean distance and average linkage in Perseus. For the calyculin A treatment, the 0-min and 30-min and the 0-min and 70-min time points were compared. p-values were calculated using a two-tailed Student's t-test assuming unequal variance. Average phosphorylation abundances of phosphorylation site upon 0-, 30-, and 70-min calyculin A treatment were compared between prophase I and other phases by 2x3 Fisher's exact test.

For GO analysis of proteins with significant changes in abundance during oocyte-to-embryo transition, human homologs of *P. miniata* proteins were identified using NCBI's BLAST+. A custom R script was then used to filter the BLAST results to only high-confidence matches (E-value<0.01) and to remove redundant matches (https://github.com/BrennanMcEwan/starfish-phos-pub-code; copy archived at *McEwan, 2021*). The resulting list was analyzed using WebGestaltR to find enriched GO terms (*Liao et al., 2019*). Motif analysis for selected and deselected amino acids surrounding phosphorylation sites in each cluster was performed using the Icelogo web interface using single localized

phosphorylation sites. Phosphorylation sites in each cluster (experimental set) were compared to all single localized phosphorylation sites maximally phosphorylated in MI (reference set). Statistical analysis to determine population difference of dephosphorylation preferences at the MI/MII transition was performed using an unpaired nonparametric Kolmogorov-Smirnov (KS) t-test. RVxF motifs based on the definition [KRL][KRSTAMVHN][VI] ⟨FIMYDP⟩[FW] ⟨ indicates excluded) with an S or T at the 'x' position or within two amino acids before and after the motif that were phosphorylated were identified in the phosphoproteomics time-course dataset and the average and standard deviation of the phosphorylation abundance were plotted.

Graphing and statistical analyses involving the scoring of meiotic phenotypes (e.g., *Figure 5E*) were performed using Prism (GraphPad). The statistical tests used to calculate p-values are indicated in the figure legends. Pixel intensity quantifications of INCENP fluorescence images (i.e., *Figure 6D*) were conducted using Fiji. Circular regions of interest (ROIs) were placed over inner centromeres or the central spindle (7 px and 30 px diameter, respectively), along with an adjacent ROI to measure the background intensity of equivalent size by RawIntDen. The background measurements were then subtracted from the centromere or midzone measurement, and the resulting value was normalized to the selection area to allow comparison between inner centromere and midzone intensity. The crystal structure of the PP2A-B55 holoenzyme (*Figure 5C*) was displayed using UCSF ChimeraX (*Goddard et al., 2018*).

## Acknowledgements

SZS was supported by a K99 fellowship from NIH/NICHD (5K99HD099315). This work was supported by grants from the NIH/National Institute of General Medical Sciences to IMC (R35GM126930) and ANK (R35GM119455), and grants to IMC from the Gordon and Betty Moore Foundation, and a Pilot award from the Global Consortium for Reproductive Longevity and Equity (GCRLE-1220). The Orbitrap Fusion Tribrid mass spectrometer was acquired with support from NIH (S10-OD016212). Molecular graphics and analyses were performed with UCSF ChimeraX, developed by the Resource for Biocomputing, Visualization, and Informatics at the University of California, San Francisco, with support from National Institutes of Health R01-GM129325 and the Office of Cyber Infrastructure and Computational Biology, National Institute of Allergy and Infectious Diseases. We thank Brooke Brauer for reading the manuscript. We thank Dr. Wolfgang Peti for purified PP1 and Dr. Richard Honkanen for tautomycetin. The RVp[S/T]F antibody was a gift from Dr. Greg Moorhead. We thank Takeo Kishimoto and Ei-Ichi Okumura for valuable comments on the manuscript and for sharing antibodies.

## Additional information

### Funding

| Funder | Grant reference number | Author |
| --- | --- | --- |
| National Institute of General Medical Sciences | R35GM126930 | Iain M Cheeseman |
| National Institute of General Medical Sciences | R35GM119455 | Arminja N Kettenbach |
| Eunice Kennedy Shriver National Institute of Child Health and Human Development | K99HD099315 | S Zachary Swartz |
| Gordon and Betty Moore Foundation | | Iain M Cheeseman |
| Buck Institute | GCRLE-1220 | Iain M Cheeseman |
| National Institute of Child Health and Human Development | 5K99HD099315 | S Zachary Swartz |

The funders had no role in study design, data collection and interpretation, or the decision to submit the work for publication.

## Author contributions
S Zachary Swartz, Conceptualization, Data curation, Formal analysis, Funding acquisition, Validation, Investigation, Writing - original draft, Writing - review and editing; Hieu T Nguyen, Data curation, Formal analysis, Investigation, Writing - review and editing; Brennan C McEwan, Data curation, Formal analysis, Investigation; Mark E Adamo, Software, Formal analysis; Iain M Cheeseman, Arminja N Kettenbach, Conceptualization, Data curation, Supervision, Funding acquisition, Project administration, Writing - review and editing

## Author ORCIDs
S Zachary Swartz https://orcid.org/0000-0002-3264-6880
Iain M Cheeseman https://orcid.org/0000-0002-3829-5612
Arminja N Kettenbach https://orcid.org/0000-0003-3979-4576

## Decision letter and Author response
Decision letter https://doi.org/10.7554/eLife.70588.sa1
Author response https://doi.org/10.7554/eLife.70588.sa2

# Additional files

## Supplementary files
• Supplementary file 1. Protein abundances quantified across time-course proteomics from prophase I arrest to the first embryonic cleavage.

• Supplementary file 2. Results of gene ontology analysis of proteins with significant changes in abundances from prophase I arrest to the first embryonic cleavage. BP - biological processes, CC - cellular components, MF – molecular function.

• Supplementary file 3. Proteomics results of oocytes treated with emetine revealing translationally regulated factors.

• Supplementary file 4. Results of gene ontology analysis of proteins affected by emetine treatment. BP - biological processes, CC - cellular components, MF – molecular function.

• Supplementary file 5. Phosphorylation abundances quantified across time-course proteomics from prophase I arrest to the first embryonic cleavage and upon calyculin A treatment.

• Transparent reporting form

## Data availability
Raw MS data for the experiments performed in this study are available at MassIVE and ProteomeXchange, accession number: PXD020916. Plasmids generated from this study are deposited to Addgene. Custom R script is available at Github (https://github.com/BrennanMcEwan/starfish-phos-pub-code; copy archived at https://archive.softwareheritage.org/swh:1:rev:7d81808b1697cf470dcd1127725e8a94c8752659).

The following dataset was generated:

| Author(s) | Year | Dataset title | Dataset URL | Database and Identifier |
|---|---|---|---|---|
| Swartz SZ, Nguyen HT, McEwan BC, Adamo ME, Cheeseman IM, Kettenbach AN | 2020 | Selective dephosphorylation by PP2A-B55 directs the meiosis I - meiosis II transition in oocytes | http://proteomecentral.proteomexchange.org/cgi/GetDataset?ID=PXD020916 | ProteomeXchange, PXD020916 |

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
