## [Decision Letter]

**Acceptance summary:**

This study undertakes a comprehensive proteomics/phosphoproteomics analysis of meiosis and the beginning of embryogenesis in the sea star, Pateria miniata. This dataset should be a valuable resource for the community. The study focuses on the regulatory circuits that drive the oocyte to embryo transition and identifies a divergent pattern of dephosphorylations consistent with an increased activity of the phosphatase PP2A-B55 that prefers to dephosphorylate phospho-threonine over phospho-serine. The differential phosphorylation observed provides a novel angle that may provide the framework for a better understanding of the steady states stabilizing meiosis I and meiosis II in the oocytes.

**Decision letter after peer review:**

[Editors’ note: the authors submitted for reconsideration following the decision after peer review. What follows is the decision letter after the first round of review.]

Thank you for submitting your work entitled "Selective dephosphorylation by PP2A-B55 directs the meiosis I – meiosis II transition in oocytes" for consideration by *eLife*. Your article has been reviewed by 2 peer reviewers, and the evaluation has been overseen by a Reviewing Editor and a Senior Editor. The following individual involved in review of your submission have agreed to reveal their identity: Peter Lenart (Reviewer #1).

Our decision has been reached after consultation between the reviewers. Based on these discussions and the individual reviews below, we regret to inform you that your work will not be considered further for publication in *eLife*.

The referees agreed on the quality of your data, and that this will be a useful resource for the community. The main concern was that a similar data set has already been published for *Xenopus* meiosis and that the conclusions from your study do not provide the insights that would warrant publication in a general audience publication such as *eLife*. If, however, you are able to address the substantive questions raised by Reviewer 2 – in particular, validating the role of NIPP1 in vivo – then we would be prepared to consider a further submission, but we judge that this will take more than two months and that the fairest decision is to return the paper to you now for you to consider your next step.

*Reviewer #1:*

The manuscript presents the, to my knowledge, first complete 'proteomics time course' through oocyte meiotic divisions up to the first embryo cleavage. This is made possible by taking advantage of the uniquely suited model system, starfish oocytes combined with state-of-the-art quantitative TMT (tandem mass tag) phosphoproteomics. The data is high quality and is a very valuable resource for the oocyte community with potential relevance for improving assisted reproduction procedures. Analysis of the data confirms several known mechanisms of phospho-regulation, and reveals new mechanisms; in particular, the authors show that the inherent phospho-threonine preference of PP2A-B55 phosphatase plays a role in differential regulation of first and second meiotic divisions.

Starfish oocytes are uniquely suited for the study of meiosis, because oocytes are available in large quantities and proceed through meiosis synchronously upon addition of the maturation hormone, and if fertilized, even continue into embryonic development. The authors took advantage of this system to carry out a 'proteomics time course': they have taken samples at 6 time points and analyzed them in a multiplexed TMT phosphoproteomics experiment. They have carefully validated the specific stages by immunostaining and microscopic analysis of oocytes from each time point.

The data appears overall high quality with over 4600 proteins and over 10 000 phosphopeptides identified in 3 replicates, comparable to studies in other species (e.g. on the range of 6000 proteins identified in *Xenopus oocytes*). This renders this dataset, which is made publicly available in its entirety, an extremely valuable resource to the community.

The data reveals an overall rather constant protein composition over the entire process of two meiotic and one mitotic cleavage division, with less than 1% of the proteins showing more than a 2-fold change. This was assumed to be the case before, and shown here for the first time. The authors validate a couple of the newly translated proteins, and show that -- consistent with earlier findings in starfish and other species -- the second meiotic M-phase require production of cyclin B protein, whereas the first mitotic cleavage division requires both cyclin A and B synthesis.

Phosphoproteomics reveals massive waves of phosphorylation and dephosphorylation, again consistent with seminal earlier studies, for example using in vivo P32 isotope labeling. The authors identify many of the previously know activating/inactivating phosphosites on regulatory proteins such as cdk1, Greatwall, ARPP19 and others. These show that before meiotic entry the high activity of phosphatases keeps phosphorylation levels low. Most excitingly, the data show that PP2A-B55 is reactivated in the MI/MII transition and drives a wave of dephosphorylation specifically affecting threonine residues, which is confirmed experimentally by overexpressing dominant negative forms of B55. Intriguingly, this preference of PP2A-B55 to TP (threonine) vs SP (serine) sites appears to have key regulatory importance. To this end, the authors show using INCENP as a model substrate that swapping threonine to serine residue indeed affects the timing of meiotic progression (that in the case of INCENP is timely relocalization from the centromeres to the spindle midzone).

Taken together, the manuscript is (i) a highly valuable resource for the community; (ii) it directly confirms several earlier indirect observations about phosphoregulation of meiotic progression; (iii) as a new result, it reveals the key importance of PP2A-B55 in meiosis-specific regulation of the cell cycle.

Overall, I am happy to recommend the manuscript for publication with minor changes:

1. I find it very important to validate the quality and synchronicity of samples. Therefore, it would be important to state the number of oocytes analyzed for Figure S1B, and also more clearly state whether these samples have been taken from the same starfish animal and season.

2. I greatly appreciate that the authors are describing the methods in detail and provide the raw proteomics data for download. Just one more detail: could they provide the MS1 and MS2 errors?

3. I would be curious to see the localization of INCENP also in MII and AnaII. Could the authors provide these additional immunofluorescence data?

*Reviewer #2:*

Swartz et al. applies MS analyses to investigate protein abundance and modification during maturation of Patiria miniata oocytes. MS analyses revealed that 99% of the app. 5000 proteins reproducibly identified displayed a max fold change of less than a 1.2-fold change indicating that meiotic cell cycle transitions are not associated with bulk changes in protein levels. Using Emetine, the authors show that progression to MII requires protein synthesis, in particular synthesis of cyclin A and B. Phospho-proteome analyses revealed that the majority of phosphorylation events occur at serine residues, followed by threonine and tyrosine residues. Prophase-I arrested oocytes displayed the lowest overall phosphorylation state. Consistent with what is known from other organisms, inhibitory phosphorylations of Cdk were found in Prophase-I arrested oocytes, but not during oocyte maturation. Then, the authors focus on PP1 and PP2A. Calyculin A treatment induced spontaneous GVBD. However, treated oocytes did not progress beyond a GVBD-like state. Cluster analyses identified TP sites as being more likely to be efficiently dephosphorylated upon exit from MI compared to SP sites. Consistent with studies in other organisms, activities of Cdk and B55 are inversely regulated such that either Cdk or B55 are active. Expression of B55 charge-swap mutations induce severe phenotypes at the MI/MII transition. T to S replacement in INCEP prevented the relocalization of the GFP-tagged fusion protein from centromeres to the spindle midzone at anaphase I.

The decision on this manuscript is not an easy one. On the one hand, the authors have done a lot of experiments and the collected data are of high quality. On the other hand, the provided insight are not particular novel or exciting and often the authors leave interesting insights without further following up. It does not come to a surprise that meiotic progression in Patiria miniata is not mediated by bulk protein turnover, but by few changes in key cell cycle regulators such as cyclins (Figures 1 and 2). Likewise, it is not surprising that phosphorylation plays a major role in meiotic cell cycle progression and that changes in Cdk and PP2A-B55 activity mediate those (Figure 3).

Given this mixed overall impression – not much new insights and some loose ends of potentially interesting insights – I think that the manuscript is more suitable for a more specialized journal rather than *eLife*. Apart from the aforementioned shortcomings, I have the following concerns.

– The authors show by Emetine treatment that protein synthesis is required for meiotic progression, but apart from cyclin B the authors don´t provide a correlation to their MS data on changes in protein abundance. Are there other interesting (cell cycle-related) candidates among the strongly upregulated proteins in the MS data that could be required for meiotic progression?

– The authors provide many indirect, but no direct evidence that PP1 activity is indeed high in Prophase I. They investigate phospho-regulation of NIPP1 in vitro without translating these insights to the oocyte. How meaningful / physiological relevant are these in vitro insights for the Prophase I arrest. Additionally, the analysis is focused on a single PP1 inhibitor, NIPP1, but PP1 activity could be regulated by several other inhibitors as well. Are there data on the phosphorylation of other known PP1 inhibitors, e.g. Inhibitor-1 or Inhibitor-2, in the MS experiments.

– Figure 4C: Do the three clusters analyzed here comprise all phosphorylated sites that have their maximum in MI? Which cluster would then contain most SP sites? Cluster 1 and 3 are depleted of Proline in the +1 position and Cluster 2 is depleted of phosphorylated Ser.

– Figure 4G: the dephosphorylation pattern for the SP and the TP sites are very similar here. It would be helpful to have a quantification of several biological replicates of this experiment to judge if there is a significant difference. Additionally, any difference might not be just to the Ser/Thr identity, because the antibodies also have a different sensitivity for adjacent amino acids.

– Does the expression of the B55 charge-swap mutants cause spontaneous GVBD? Importantly, with this mutant in hand, the authors could investigate by WB if known B55 substrates show altered dephosphorylation kinetics.

– Figure 6A, the authors want to make the point that phosphorylation of T61 sharply decreased following MI. S1158 shows an even sharper decline in its phosphorylation level, while other phosphorylated T sites seem not to be dephosphorylated, e.g. T506, indicating that the dephosphorylation code is more complex than being encoded in the nature of the phosphorylated residue. Why did the authors not investigate the phosphorylation state of T61 in their charge-swap experiment?

– In their analysis the authors talk a lot about what is happening during the transition from MI to MII, although their MS data just provide information about metaphase of MI and metaphase of MII, but not for the situation in between. In the data from previous publications (e.g. Okano-Uchida et al., 1998) it seems as if Cdk1 activity is very high in MI, then drops almost completely between MI and MII before rising again towards metaphase of MII (although not as high as in MI). So, in theory, it could be that there is much more B55-dependent dephosphorylation happening between MI and MII than was measured here, but a specific subset of sites (eg SP sites) got preferentially rephosphorylated for MII (e.g. defined by different Cdk activity thresholds). The INCENP data suggest that Thr sites are earlier dephosphorylated than Ser sites, but it might just be a matter of timing and not if a site is at all dephosphorylated or not between MI and MII as suggested here.

[Editors’ note: further revisions were suggested prior to acceptance, as described below.]

Thank you for resubmitting your work entitled "Selective dephosphorylation by PP2A-B55 directs the meiosis I – meiosis II transition in oocytes" for further consideration by *eLife*. Your revised article has been evaluated by Anna Akhmanova (Senior Editor) and a Reviewing Editor.

All three reviewers recommend acceptance of your paper but only after the inclusion of controls for the morpholine experiments (see Referee 3, point 3), and some extensive re-writing.

In your rewriting, the referees would like you to place your confirmatory data in proper context (eg: PMID: 29643273 and note referee 3, points 1 and 2), and mine your data further (referee 3 points 6 and 7). These results should then be discussed with reference to other systems (referee 3 point 8).

*Reviewer #2:*

As before, I remain supportive of the publication of the revised manuscript. Indeed, the authors carried out additional experiments and added data and explanations that together significantly strengthened their conclusions, and in my opinion these addressed most of the criticism raised by me and Reviewer 2.

Specifically, the added data for translationally regulated proteins is another important resource for the community. More importantly, the authors performed additional experiments to demonstrate the major role of PP2A-B55 in meiotic regulation, in contrast to PP1 having a minor contribution. Further, the authors provide additional Western blots using phospho-specific antibodies in support of their conclusion of differential TP/SP phosphorylation in MII.

Together, besides providing a very useful resource to the community, the authors present a conceptually new mode of meiotic regulation based on the preferential TP vs SP dephosphorylation by PP2A-B55 required for successful completion of the second meiotic division.

*Reviewer #3:*

This study focuses on the phosphoproteomics analysis during the two specialized meiotic divisions and beginning of embryogenesis in the sea star Pateria miniata. Dr. Swartz and collaborators have used a comprehensive proteomics/phosphoproteomics approach to investigate the regulatory circuits that drive the oocyte to embryo transition in this species. The first part of the study includes largely confirmatory observations of widely accepted concepts on regulation of the meiotic cell cycle. In the second part, the authors identify a divergent pattern of dephosphorylations consistent with an increased activity of the phosphatase PP2A-B55. Even though a PP2A role at this transition has been reported, the differential phosphorylation observed provides a novel angle that may provide the framework for a better understanding of the steady states stabilizing MI and MII in the oocytes.

The authors should be commended for the large amount of data they have generated and that certainly will be a useful resource for the meiosis/cell cycle community. The authors have responded to the previous review in a constructive way by including new data in support of their conclusions. Unfortunately and instead of using this large wealth of data to explore new features of meiosis or settle the numerous outstanding issues, they spend a good part of the study for largely confirmatory experiments. Nevertheless, the observation of the global differential dephosphorylation of TP and SP sites would be an interesting finding worth reporting (even though the PP2A- B55 preferential dephosphorylation of TP sites has already been described). Some missing controls would strengthen the authors' conclusions.

1. By using a proteomic approach the authors conclude that protein expression does not change substantially throughout the entire sea star cell cycle, which is already well established (see PMID: 29643273). However, it is also established that a number of critical proteins are synthesized during these transitions in the sea star. In addition to cyclin B and cyclin A and Pim-1 mentioned by the authors, Mos, cyclin E, Cdk2 and Wee1 accumulate during the two-cell cycle and pronuclear stages. The authors should report whether their proteomics approach confirms these previous findings. This would confirm the quality of the proteomics data collected. A similar question had already been posed during the previous review. In the discussion, it would be useful to comment that the stability of the proteome associated with minimal new protein synthesis is a peculiarity of the sea star, as large changes in protein synthesis drive the meiotic division in other animal models like the frog and mammals.

2. Page 5-12 of the manuscript report experiments that explore basic changes in phosphorylation and the involvement of phosphatases and essentially confirm widely established concepts for sea star meiosis and are consistent with basic views established for most species studied. All these properties have been widely described and well summarized, for instance in PMID: 29643273. In this comprehensive review, Figure 3 is used to summarize all these temporal changes in activity driving meiosis; it is hard to discern how this section of the manuscript adds any significant new information. This is puzzling because the data collected by the authors could have been mined to provide substantial and more relevant information and they have not used the data generated to resolve several contentious issues. For instance, the authors have the phosphorylation data for the prophase1 to GVBD transition to distinguish the state of phosphorylation of Myt 1 and Cdc25 at Cdk1 and SGK sites, not to mention the PI-3K dependent phosphorylation sites of SGK. There are reports suggesting differences, albeit subtle, in the consensus AKT/SGK sites. This analysis would have helped to consolidate the recent reports that SGK, and not Akt, functions distally to 1-methyladenine receptor. Moreover, the data could have been used to better tease apart at GVBD/MI what is usually called "initial activation" versus "auto-amplification" components required for switch/like maximal MPF activity. The authors could have reported and discussed the dual ARPP19 phosphorylations by Greatwall and CDK1.

3. Page 7. The authors use morpholino oligonucleotides to block the translation of cyclin A and B. However, they provide no control data on the extent and selectivity of the knockdown. This control is necessary to draw any conclusion.

4. Results, page 13. In the revised manuscript, the authors have included additional data with a more informative sampling rate between MI and MII to include points at anaphase. Using TP and SP specific antibodies the authors confirm that dephosphorylation at the TP site has faster kinetics than dephosphorylation at SP sites. The authors conclude that "despite the transient inactivation of Cdk1 between the meiotic division, SP phosphorylation remains relatively stable during this window". They use this finding to rule out the possibility of a kinase rephosphorylating at the SP sites. However, the authors do not discuss the possibility that the divergent MAPK activity, which remains constantly high, while Cdk1 activity is declining, at least in part contributes to divergent SP/TP phosphorylation state. This should be further discussed and possibly addressed experimentally.

5. To disrupt PP2A-B55 function, the authors use a dominant-negative approach with point mutations in B55 that disrupt interactions with substrates. The authors provide some data on the effect on meiotic progression. However, they do not investigate whether disruption of PP2A-B55 anchoring indeed blocks the divergent TP/SP dephosphorylations. This could be explored using phospho-specific antibodies.

6. The authors impute the divergent dephosphorylation of TP and SP sites to the selectivity of PP2A-B55. If this were correct, one should find a mirror image of dephosphorylation observed at the MI/MII transition during prophase to GVBD transition. PP2A is active in prophase 1, but it becomes inactivated at GVBD/MII. One should then see a preferential dephosphorylation of TP sites in prophase with little change in the SP sites. The authors should discuss this possibility and provide the data they have available; a hint of this is reported in Figure 6A.

7. INCEMP is a highly phosphorylated protein with a large number of sites to choose from. The authors report changes in the TP and SP sites, which clearly diverge. This is considered a major strength of the study, as it verifies that spatial sequestration of the protein is not an issue in the differential TP/SP dephosphorylation. The authors should include the SP phosphorylation patterns also for PRC1 and TPX2. Now only TP patterns are reported in Figure 5-S3. This would confirm that the TP/SP divergent dephosphorylation is of wide applicability.

8. In the discussion the authors make no attempt to compare and contrast their data with the wealth of information reported for other species. The authors do not specify the species used in the title. In the same vein, in the opening statement of the discussion the authors claim to have defined an extensive program of phosphorylation during the oocyte-to-embryo transition. However, they do not mention that they do so in the sea star. This has already been reported for other species. In addition, the authors do not discuss that the sea star provides a simplified experimental model to investigate regulation of meiosis and that many additional layers of regulation are present in frogs and mammals. Thus, the mechanisms of meiotic resumption vary considerably among species including sea star, frog and mouse. This important concept is not captured in the discussion.

*Reviewer #4:*

As outlined in my comments on the initial manuscript, I was not sure if the novelty of the findings suffice publications in *eLife*. For the revisions, the authors have done a great job to address all the points I raised. The manuscript significantly improved and I therefore recommend publication.

---

## [Author Response]

[Editors’ note: the authors resubmitted a revised version of the paper for consideration. What follows is the authors’ response to the first round of review.]

The referees agreed on the quality of your data, and that this will be a useful resource for the community. The main concern was that a similar data set has already been published for Xenopus meiosis and that the conclusions from your study do not provide the insights that would warrant publication in a general audience publication such as eLife. If, however, you are able to address the substantive questions raised by Reviewer 2 – in particular, validating the role of NIPP1 in vivo – then we would be prepared to consider a further submission, but we judge that this will take more than two months and that the fairest decision is to return the paper to you now for you to consider your next step.

We thank the reviewers for their evaluation of our work. Importantly, we believe that our study represents a substantial advance from prior proteomic studies in oocytes. In particular, our work provides improved temporal resolution over these prior studies by unambiguously correlating cell cycle state to the phosphorylation landscape of the oocyte, as well as testing time points that span the entire oocyte-to-embryo transition. To accomplish this, we leveraged the unique biology of the sea star, which provides excellent synchrony and cell biological tractability. We have carefully cited these prior studies and provided a comparison to our work, but it is important to highlight that we were unable to do this clearly in many cases due to incomplete time courses, missing information, and ambiguity in cell cycle stages. Furthermore, Presler et al. PNAS 2017 only reported ~3500 phosphorylation events. Peuchen et al. Scientific Reports 2017 reported 8971 phosphorylation events identified across the two biological and technical replicates that they performed, but with only limited statistical analysis to assess the reproducibility of the data (the authors comment that “For the phosphoproteome experiments, biological duplicates and technical duplicates provided reasonable correlation (Pearson coefficient > 0.7)”). For our time course analyses, we identify a total of 25,227 phosphorylation sites across three biological replicates, of which 16,691 are identified in two or more replicates. The average Pearson coefficient of our dataset is 0.84 and 12,481 sites are reproducibly quantified (p-value 0.05 or less).

Thus, we believe that our proteomics analysis provides a definitive picture of the phosphorylation landscape across the entire oocyte-to-embryo transition. In addition to the value of our study as a resource for future work, our work suggests a new paradigm by which the MI / MII transition is regulated through differential serine vs. threonine conservation, and a critical role for the phosphatase PP2A-B55. To our knowledge, this has not been presented before and we believe this represents an important advance.

In our revised manuscript, we provide further support for the central role of PP2A-B55 as a conductor of meiotic prophase I arrest, and the transition between the two meiotic divisions. In response to the reviewer comments and discussion, we have undertaken a substantial effort to further test the role of PP1. Through proteomic and cell biological analysis, we find that PP1 plays a modest role relative to PP2A-B55 in the developmental transitions tested in our study. We believe these data substantially improve the manuscript by providing deeper insights into the relative contributions of these phosphatases.

Reviewer #1:[…]1. I find it very important to validate the quality and synchronicity of samples. Therefore, it would be important to state the number of oocytes analyzed for Figure S1B, and also more clearly state whether these samples have been taken from the same starfish animal and season.

Thank you for raising this important point. We have now included the number of oocytes analyzed in the legend for Figure 1—figure supplement 1B (“each time point represents at least 49 oocytes”). To better document the synchrony, we have added the following to the methods section: “Oocyte samples for proteomics were collected from a single female within the same season to maximize synchrony between time courses. Stereotypical meiotic timings were visually confirmed in live oocytes using germinal vesicle breakdown and polar body emission as metrics, before snap freezing samples in liquid nitrogen. The synchrony data in Figure S1B were performed with oocytes from a second animal collected at the same season and location that was confirmed to undergo meiosis at a similar rate.”

2. I greatly appreciate that the authors are describing the methods in detail and provide the raw proteomics data for download. Just one more detail: could they provide the MS1 and MS2 errors?

We are pleased that the methods section and raw data were helpful. We have now included the additional requested details. We have adjusted the methods section to include additional information on the data filtering:

“High resolution tandem mass spectra were searched using COMET with a precursor ion tolerance of +/- 1 Dalton and a fragment ion tolerance of +/- 0.02 m/z, static mass of 229.162932 on peptide N-termini and lysines and 57.02146 Da on cysteines, and a variable mass of 15.99491 Da on methionines against a target-decoy version of the Patiria miniata proteome sequence database. For phosphoproteomics analysis 79.96633 Da on serines, threonines and tyrosines was included as an additional variable mass. The resulting peptide spectral matches (PSMs) were filtered to a < 1% false discovery rate (FDR) using the target decoy strategy with a typical precursor ion mass filter of +/- 1.5 parts-per-million (PPM) mass accuracy and corresponding XCorr and dCn values. Quantification of LC-MS/MS spectra was performed using in house developed software. Probability of phosphorylation site localization was determined by PhosphoRS (Taus et al., 2011).”

3. I would be curious to see the localization of INCENP also in MII and AnaII. Could the authors provide these additional immunofluorescence data?

Thank you for this suggestion. Based on this comment and related suggestions from Reviewer #2, we have now conducted multiple additional experiments to analyze the role of PP2A-B55 and differential dephosphorylation in controlling INCENP localization. First, we have analyzed the localization for wild type INCENP across Meiosis I and Meiosis II. The localization behavior for INCENP in Meiosis II is now included in Figure 5—figure supplement 3D. In contrast to Meiosis I, we detect reduced INCENP localization to centromeres in Meiosis II, but INCENP additionally localizes to the Anaphase II midzone, and nucleolar structures in the female pronucleus. Second, we tested INCENP localization in oocytes expressing the inhibitory B55 mutants that alter its substrate specificity. Under these conditions, INCENP localization to metaphase centromeres is increased, and the translocation to the midzone in anaphase is strongly reduced. These data are now included in Figure 6. This localization is strongly consistent with our model that the selective dephosphorylation of INCENP T61 in Meiosis II by B55 governs the localization of INCENP to the centromere versus the midzone.

Reviewer #2:Swartz et al. applies MS analyses to investigate protein abundance and modification during maturation of Patiria miniata oocytes. MS analyses revealed that 99% of the app. 5000 proteins reproducibly identified displayed a max fold change of less than a 1.2-fold change indicating that meiotic cell cycle transitions are not associated with bulk changes in protein levels. Using Emetine, the authors show that progression to MII requires protein synthesis, in particular synthesis of cyclin A and B. Phospho-proteome analyses revealed that the majority of phosphorylation events occur at serine residues, followed by threonine and tyrosine residues. Prophase-I arrested oocytes displayed the lowest overall phosphorylation state. Consistent with what is known from other organisms, inhibitory phosphorylations of Cdk were found in Prophase-I arrested oocytes, but not during oocyte maturation. Then, the authors focus on PP1 and PP2A. Calyculin A treatment induced spontaneous GVBD. However, treated oocytes did not progress beyond a GVBD-like state. Cluster analyses identified TP sites as being more likely to be efficiently dephosphorylated upon exit from MI compared to SP sites. Consistent with studies in other organisms, activities of Cdk and B55 are inversely regulated such that either Cdk or B55 are active. Expression of B55 charge-swap mutations induce severe phenotypes at the MI/MII transition. T to S replacement in INCEP prevented the relocalization of the GFP-tagged fusion protein from centromeres to the spindle midzone at anaphase I.The decision on this manuscript is not an easy one. On the one hand, the authors have done a lot of experiments and the collected data are of high quality. On the other hand, the provided insight are not particular novel or exciting and often the authors leave interesting insights without further following up. It does not come to a surprise that meiotic progression in Patiria miniata is not mediated by bulk protein turnover, but by few changes in key cell cycle regulators such as cyclins (Figures 1 and 2). Likewise, it is not surprising that phosphorylation plays a major role in meiotic cell cycle progression and that changes in Cdk and PP2A-B55 activity mediate those (Figure 3).

We appreciate this feedback on the quality of our data and the questions of novelty. We agree that it comes as no surprise that phosphorylation is an important driver of cell cycle progression building on prior work across multiple organisms. However, the differential dephosphorylation of threonine vs. serine as a paradigm to drive the MI / MII transition has not been proposed to our knowledge. Our work provides answers to critical questions including how oocytes complete Meiosis I, and then proceed directly in to Meiosis II without undergoing a gap phase or DNA replication, and how Meiosis I-specific functions are reversed while maintaining an overall meiotic state. The differential behavior of TP versus SP phosphorylation, mediated by PP2A-B55, provides an important, new conceptual framework to understand this process – findings that establish this paradigm and create new directions for future work in this area. In addition, we believe our study provides a uniquely holistic view of the oocyte-to-embryo transition, enabled by the powerful experimental advantages of the sea star. Together, we feel that our work provide a substantial conceptual advance, as well as a valuable resource that will be broadly useful for researchers interested in exploring this specialized cell cycle program.

– The authors show by Emetine treatment that protein synthesis is required for meiotic progression, but apart from cyclin B the authors don´t provide a correlation to their MS data on changes in protein abundance. Are there other interesting (cell cycle-related) candidates among the strongly upregulated proteins in the MS data that could be required for meiotic progression?

Thank you for this helpful point. We have now addressed this in two ways. First, we now specifically highlight additional proteins whose steady-state levels fluctuate in our time-course proteomics. This includes proteins such as the kinase PIM-1 and Importin 8, which are translated de novo after meiotic resumption. Second, we have now conducted additional mass spectrometry experiments in which we compare control Meiosis II oocytes to those in which new translation was blocked with emetine. With this approach, we identified 108 proteins whose levels are sensitive to translational inhibition. These factors include PIM-1, as well as the DNA replication factor Cdt1, a potential Cdk regulatory subunit, and other proteins that are now discussed in the Results and Discussion sections. We include these data as an additional supplemental figure (Figure 2—figure supplement 1) and excel sheet (Supplementary file 3). We believe that this dataset provides a valuable resource and additional useful insights into the proteins for which nascent translation is necessary for meiotic progression.

– The authors provide many indirect, but no direct evidence that PP1 activity is indeed high in Prophase I. They investigate phospho-regulation of NIPP1 in vitro without translating these insights to the oocyte. How meaningful / physiological relevant are these in vitro insights for the Prophase I arrest. Additionally, the analysis is focused on a single PP1 inhibitor, NIPP1, but PP1 activity could be regulated by several other inhibitors as well. Are there data on the phosphorylation of other known PP1 inhibitors, e.g. Inhibitor-1 or Inhibitor-2, in the MS experiments.

Thank you for this interesting question regarding the physiological contribution of PP1 and its inhibitory partners. This comment prompted us to undertake a substantial investigation of PP1 behavior by mass spectrometry, Western blotting, and in vivo experiments in oocytes. First, in addition to our previous analysis of NIPP1, we identified a sea star ortholog of the PP1 regulatory protein Inhibitor-2 (I2). We find that I2 also becomes phosphorylated on several sites following hormonal stimulation and meiotic resumption. These phosphorylation sites could regulate the inhibitory activity of I2 (Author response image 1). However, we generated both wild-type and phosphomimetic mutations for these sites, and found no dominant effect when the constructs were expressed in oocytes.

**Author response image 1. sa2fig1:** IPP2 phosphorylation sites.

Second, we analyzed our phosphoproteomic data for a well-characterized PP1 binding motif, RVxF (in which “x” is S or T). Phosphorylation of these motifs by Aurora and other kinases negatively regulates the recruitment of PP1 to its regulatory partners and substrates. We find that RVxF phosphorylation is low in Prophase I, but increases in meiosis, consistent with a corresponding downregulation of PP1 activity. We further tested this trend by Western blotting with antibodies specific to the RVp[S/T]F motif, which revealed consistent results. We have included these data in Figure 3—figure supplement 2C, and figure supplement 3.

Having defined these temporal phosphorylation behaviors indicative of PP1 activity across meiosis, we next generated multiple constructs to evaluate the in vivo contributions of PP1 dephosphorylation in oocytes. Under the same conditions in which we tested the PP2A-B55 mutant constructs and the inhibitor Calyculin A, we evaluated the effects of expressing the following constructs in oocytes: 1) Wild-type NIPP1, 2) the NIPP1 S197,199A mutant, 3) wildtype I2, 4) a phosphomimetic I2 mutant, and 5) direct injection of RVSF peptides to compete off PP1 from endogenous substrates. Notably, none of these perturbations had a substantial effect on the maintenance of the Prophase I arrest, meiotic resumption, or the successful extrusion of polar bodies.

Finally, to directly test the role of PP1, we used the inhibitor Tautomycetin, which is highly selective for PP1 vs. PP2A. Using in vitro assays, we first confirmed that Tautomycetin is indeed a highly potent and selective inhibitor of PP1. Next, we tested whether Tautomycetin could induce spontaneous germinal vesicle breakdown, using the same assay as we previously conducted with Calyculin A. Strikingly, we found no effect even after a 24 hour treatment with this inhibitor under identical conditions as those used for Calyculin A. Finally, we found that Tautomycetin-treated oocytes progressed through meiosis with normal kinetics including undergoing polar body extrusion. However, Tautomycetin treatment did result in significant chromosome alignment and mis-segregation defects, consistent with an inability to correct kinetochore-microtubule attachment errors. These data have been added to Figure 3—figure supplement 6.

In summary, based on the reviewer’s valuable suggestion, we have now monitored and perturbed the activity of PP1 and its inhibitory partners using multiple orthogonal approaches. This analysis allows us to now conclude that PP1 plays a comparatively modest role in maintaining a Prophase I arrest and driving meiotic progression relative to PP2A. We feel that this data provides a substantial conceptual advance for the paper, and further emphasizes the central function of PP2A-B55 in orchestrating meiosis.

– Figure 4C: Do the three clusters analyzed here comprise all phosphorylated sites that have their maximum in MI? Which cluster would then contain most SP sites? Cluster 1 and 3 are depleted of Proline in the +1 position and Cluster 2 is depleted of phosphorylated Ser.

For the motif analysis in Figure 4C, we only included singly phosphorylated peptides. In addition to commenting on this in the Methods section, we have now included this information in the Results section. We made this choice because, for peptides with more than one phosphorylation site, we cannot discern which sites are responsible for the changing abundance. However, in the previous version of Figure 4A, all sites peaking in MI were clustered. For the revised paper, we now repeated the clustering analysis using only peptides with single phosphorylation sites and have adjusted Figures 4A and B accordingly. The conclusions for these new analyses are highly comparable with our previous findings.

For the analysis in Figure 4C evaluating the over-and under-representation of specific residues, each position is investigated independently and compared against the amino acid composition of all phosphorylation sites at this position. Although there are more SP than TP sites in Clusters 1 and 2, the number of proline-directed sites in these clusters is not overrepresented (Author response image 2). However, in Cluster 3, nearly all threonine phosphorylation sites are proline-directed.

**Author response image 2. sa2fig2:** Distribution of S and T residues, and SP and TP motifs in Cluster 1 3.

– Figure 4G: the dephosphorylation pattern for the SP and the TP sites are very similar here. It would be helpful to have a quantification of several biological replicates of this experiment to judge if there is a significant difference. Additionally, any difference might not be just to the Ser/Thr identity, because the antibodies also have a different sensitivity for adjacent amino acids.

Thank you for this suggestion. To increase the confidence in our time course Western blot experiments, we have now performed multiple independent experiments and present the results as the mean and standard deviation of three biological replicates. These new data are included in Figure 4G and Figure 4—figure supplement 1B, and emphasize the reproducibility of the differences we discovered in SP vs. TP behavior.

In addition, we have now tested antibodies that recognize a different phosphorylation consensus as an additional orthogonal metric. We obtained commercially-available antibodies that recognize pTPP and pSPP, and found similar temporal phosphorylation profiles as those generated with the previous pTPxK and [K/H]pSP antibodies. These data are now included in Figure 4—figure supplement 2.

– Does the expression of the B55 charge-swap mutants cause spontaneous GVBD? Importantly, with this mutant in hand, the authors could investigate by WB if known B55 substrates show altered dephosphorylation kinetics.

Thank you for pointing out this interesting possibility. We have now tested the effect of the B55 DE/A mutant on maintaining the Prophase I arrest. Similar to Calyculin A treatment, expression of this B55 mutant induces spontaneous GVBD, and results in the subsequent apoptosis of the oocytes within 3 days of culture. These data are now included in Figure 5— figure supplement 2. This indicates that PP2A-B55 perturbation alone is sufficient to phenocopy Calyculin A treatment, and supports a model in which PP2A is the primary phosphatase that acts to maintain the Prophase I arrest. We believe this is a valuable conceptual advance for the regulation of oocyte arrest.

We agree that it would be very valuable to directly test the dephosphorylation kinetics of known B55 substrates by Western blot. Unfortunately, we are technically unable to do this experiment as we currently lack antibodies that recognize specific B55 substrates in the sea star. As an alternative approach, we now report the cellular behavior of INCENP-GFP, a PP2A-B55 substrate, following expression of either wild-type or mutant B55. We find that altering the substrate specificity of PP2A with this B55 mutant substantially reduces the ability of wild-type INCENP to translocate to the central spindle in anaphase of Meiosis I. In addition, this perturbation increases the localization of INCENP to centromeres in Metaphase. These localization defects are consistent with a failure in the timely dephosphorylation of INCENP T61, based on our analysis of this mutant construct (Figure 6D,E). These data have now been added to Figure 6.

– Figure 6A, the authors want to make the point that phosphorylation of T61 sharply decreased following MI. S1158 shows an even sharper decline in its phosphorylation level, while other phosphorylated T sites seem not to be dephosphorylated, e.g. T506, indicating that the dephosphorylation code is more complex than being encoded in the nature of the phosphorylated residue. Why did the authors not investigate the phosphorylation state of T61 in their charge-swap experiment?

Thank you for pointing this out. In our discussion of the results, we did not fully capture the nuance of the phosphorylation code. It is not simply the threonine vs. serine that drives these kinetics, but likely also depends on the identity of the adjacent amino acids. In the case of T506, the presence of acidic amino acids (E508 and D510) are disfavored for B55 association, as supported by our motif analysis (Figure 4C-E). We agree that the behavior of S1158 is unique, and the reason for this difference are less obvious, but we suggest that perhaps this residue’s position at the extreme C-terminus of the protein may influence its accessibility to B55.

To better describe and explain these behaviors, we have added several lines to the Results and Discussion section to convey that the dephosphorylation code is more complex than the single phosphorylated residue. (Results: “In contrast, T506 remains relatively stable, likely due to the presence of downstream acidic amino acids, thereby disfavoring PP2A-B55 association (Figure 6A).”, Discussion: “This phosphorylation code is further modulated by the charge of adjacent amino acids, with positively charged residues favoring PP2A-B55 association”).

Unfortunately, it is not technically possible at this time to directly evaluate the phosphorylation state of INCENP T61 following the expression of B55 mutants. We lack phospho-specific antibodies recognizing this residue, and testing this by mass spectrometry would require the injection of a prohibitive number of oocytes to achieve sufficient quantities for this analysis. As an alternative approach to this, we now report the localization behavior of INCENP-GFP following expression of mutant B55. As our model predicts, INCENP translocation to the central spindle in anaphase is strongly reduced when B55 activity is disrupted, whereas the localization to centromeres is increased. These results are consistent with the behavior governed by delayed dephosphorylation of T61.

– In their analysis the authors talk a lot about what is happening during the transition from MI to MII, although their MS data just provide information about metaphase of MI and metaphase of MII, but not for the situation in between. In the data from previous publications (e.g. Okano-Uchida et al., 1998) it seems as if Cdk1 activity is very high in MI, then drops almost completely between MI and MII before rising again towards metaphase of MII (although not as high as in MI). So, in theory, it could be that there is much more B55-dependent dephosphorylation happening between MI and MII than was measured here, but a specific subset of sites (eg SP sites) got preferentially rephosphorylated for MII (e.g. defined by different Cdk activity thresholds). The INCENP data suggest that Thr sites are earlier dephosphorylated than Ser sites, but it might just be a matter of timing and not if a site is at all dephosphorylated or not between MI and MII as suggested here.

Thank you for suggesting this interesting model. We agree that rephosphorylation of selected residues is an alternative possibility that could explain the phosphorylation kinetics that we observe. To test this, we have conducted several additional experiments. First, we have now conducted a higher temporal resolution analysis of phosphorylation behaviors across the oocyte-to-embryo transition using Western blotting with multiple pTP and pSP CDK consensus antibodies, in biological triplicate. We find that serine phosphorylation is substantially more stable between MI and MII than threonine phosphorylation (Figure 4G, Figure 4—figure supplement 1B). The increased sampling at multiple time points during the MI/MII transitions allows us to more confidently define these differences, and importantly, indicates that there is not an apparent decrease in phosphorylation followed by rephosphorylation. As the reviewer points out, this is the window of meiosis during which cyclin B is fully degraded and is prior to Cdk1 reactivation. We therefore believe that this would disfavor a model in which selective rephosphorylation of serine-proline by Cdk explains this difference in behavior. We now discuss these alternative possibilities with the following lines in the Results section: “However, an alternative interpretation is that both SP and TP sites are dephosphorylated equally at the MI/MII transition, but then SP sites are selectively and rapidly re-phosphorylated in MII. To distinguish between these models, we performed Western blots using antibodies against phosphorylated pTP and pSP CDK motifs on samples collected with increased temporal resolution…”

Finally, prior work suggests that Cdk1 preferentially phosphorylates threonine residues (Miller et al., Science Signaling, 35, 2008), further arguing against this model. This is further supported by the S/T distribution of 1576 Cdk1 substrate phosphorylation sites listed on Phosphosite.org. Both sources indicated that ~30% of Cdk1 phosphorylation sites are threonine, which is substantially higher than the ~15.5% phosphothreonine that would be expected for a non-S/T selective kinase based on the global occurrence of phosphorylated threonine residues (Sharma et al., Cell Reports, 8, 2014).

[Editors’ note: what follows is the authors’ response to the second round of review.]

Reviewer #3:This study focuses on the phosphoproteomics analysis during the two specialized meiotic divisions and beginning of embryogenesis in the sea star Pateria miniata. Dr. Swartz and collaborators have used a comprehensive proteomics/phosphoproteomics approach to investigate the regulatory circuits that drive the oocyte to embryo transition in this species. The first part of the study includes largely confirmatory observations of widely accepted concepts on regulation of the meiotic cell cycle. In the second part, the authors identify a divergent pattern of dephosphorylations consistent with an increased activity of the phosphatase PP2A-B55. Even though a PP2A role at this transition has been reported, the differential phosphorylation observed provides a novel angle that may provide the framework for a better understanding of the steady states stabilizing MI and MII in the oocytes.The authors should be commended for the large amount of data they have generated and that certainly will be a useful resource for the meiosis/cell cycle community. The authors have responded to the previous review in a constructive way by including new data in support of their conclusions. Unfortunately and instead of using this large wealth of data to explore new features of meiosis or settle the numerous outstanding issues, they spend a good part of the study for largely confirmatory experiments. Nevertheless, the observation of the global differential dephosphorylation of TP and SP sites would be an interesting finding worth reporting (even though the PP2A- B55 preferential dephosphorylation of TP sites has already been described). Some missing controls would strengthen the authors' conclusions.1. By using a proteomic approach the authors conclude that protein expression does not change substantially throughout the entire sea star cell cycle, which is already well established (see PMID: 29643273). However, it is also established that a number of critical proteins are synthesized during these transitions in the sea star. In addition to cyclin B and cyclin A and Pim-1 mentioned by the authors, Mos, cyclin E, Cdk2 and Wee1 accumulate during the two-cell cycle and pronuclear stages. The authors should report whether their proteomics approach confirms these previous findings. This would confirm the quality of the proteomics data collected. A similar question had already been posed during the previous review.

We have further highlighted in the text that our observations on protein stability are consistent with previous observations of the limited turnover of specific proteins. Because of the global scale of our analysis, we can now extend the previous observations for specific proteins to a more generalizable paradigm.

A challenge in proteomic versus genomic analyses is the dynamic range problem, particularly when combining proteomics with quantitative approaches such as tandem mass tagging. Although modern mass spectrometry instrumentation has come far, it is still not at the point where all expressed proteins are easily identified and quantified to the same extent regardless of their expression level. Although our analysis is, to the best of our knowledge, one of the most comprehensive investigations into a meiotic proteome to date, and the only one that has been conducted in the sea star, a subset of low expression level proteins are absent from this data. The abundance of many cell cycle regulators, including Mos, Cyclin E, Cdk2, and Wee1, is low and tightly controlled, and therefore often not accessible yet by proteomics approaches without additional enrichment strategies. Although we do detect phosphorylation sites on these four proteins, we were only able to identify and quantify Wee1 on the proteome level in 2 of the 3 biological replicates with a small number of peptides (3 in replicate 2, and 1 in replicate 3). The detection of phosphorylation sites is likely due to the 100-fold enrichment upon phospho-peptide purification. Although this limits our ability to make statements about these proteins, it is not a data quality problem, but rather a depth of analysis problem typical of all standard mass spectrometry approaches.

In the discussion, it would be useful to comment that the stability of the proteome associated with minimal new protein synthesis is a peculiarity of the sea star, as large changes in protein synthesis drive the meiotic division in other animal models like the frog and mammals.

We appreciate the comment, but disagree with this point. A previous analysis by Presler et al. (PNAS 2017, Figure 1 C and D) of *Xenopus oocytes* found that only 48 out of 8,641 proteins were dynamic, whereas the majority of proteins remain constant during a time course of 0 – 20 minutes after fertilization and during progression from metaphase II to anaphase II. Similarly, Peuchen et al. (Scientific Reports 2017, Figure 1B) report that there is “not a substantial change in the *Xenopus* proteome” in a time course of oocyte maturation to first zygotic cleavage. They find that only 486 of 6428 proteins change in expression by more than 5%. Based on these observations, we believe that, at least in frogs, there are also not significant changes in protein levels on a global scale. Although some selected critical regulatory proteins do change in their protein levels or undergo new synthesis (as described in this paper), thereby driving meiotic progression, the majority of proteins remain unchanged during this developmental window. It is possible that this reviewer is referring to the maternal to zygotic transition, during which time there are likely to be more substantial changes.

2. Page 5-12 of the manuscript report experiments that explore basic changes in phosphorylation and the involvement of phosphatases and essentially confirm widely established concepts for sea star meiosis and are consistent with basic views established for most species studied. All these properties have been widely described and well summarized, for instance in PMID: 29643273. In this comprehensive review, Figure 3 is used to summarize all these temporal changes in activity driving meiosis; it is hard to discern how this section of the manuscript adds any significant new information. This is puzzling because the data collected by the authors could have been mined to provide substantial and more relevant information and they have not used the data generated to resolve several contentious issues. For instance, the authors have the phosphorylation data for the prophase1 to GVBD transition to distinguish the state of phosphorylation of Myt 1 and Cdc25 at Cdk1 and SGK sites, not to mention the PI-3K dependent phosphorylation sites of SGK. There are reports suggesting differences, albeit subtle, in the consensus AKT/SGK sites. This analysis would have helped to consolidate the recent reports that SGK, and not Akt, functions distally to 1-methyladenine receptor.

We thank the reviewer for their helpful comments. As suggested, we have now further mined our data. Although we do not detect the activation loop phosphorylation site (T312) in SGK in our time course dataset, we do observe phosphorylation of the neighboring TP (T316) site, which is part of the P+1 loop and activation segment.

Concomitantly, we observe an increase in the phosphorylation of Cdc25 S188, an SGK site (Hiraoka et al. JCB, 2019), which serves to activate Cdc25. Furthermore, we observe an increase in phosphorylation of a double and triple phosphopeptide of Myt1, which includes S75, an SGK site (Hiraoka et al. JCB, 2019), which functions to inactivates Myt1. Activation of Cdc25 and inactivation of Myt1 trigger activation of Cdk1-Cyclin B (Hiraoka et al. JCB, 2019).

Based on an analysis of the known human substrates for SGK1 and Akt1 (Phosphosite.org), we propose that the most dramatic difference in the SGK1 and Akt1 consensus motifs is a preference in basophilic residues in position -4 for Akt1 (see Author response image 3).

**Author response image 3. sa2fig3:** 

Based on the reviewer’s suggestion, we have now investigated the presence of RxRxxS/T consensus motifs in the dataset for the localized, single, and reproducibly quantified phosphopeptides that increased in phosphorylation abundance by 3-fold or more from Prophase 1 to GVBD versus sites with less than a 3fold increase over the same time period.This analysis revealed a stronger preference for basic amino acids in the -4 position, indicative of Akt phosphorylation for phosphorylation sites with a minimal increase in abundance in the Prophase 1 to GVBD position. Although we find this data intriguing and suggestive of a model where there is stronger SGK activation than Akt, we do not feel that this point represents a major conclusion that merits specific highlighting in the manuscript. We hope that this information will be useful to readers who are reading this as part of the transparent review process, and also believe that this highlights the value of this broad dataset for diverse researchers interested in phospho-regulation downstream of diverse kinases.

Moreover, the data could have been used to better tease apart at GVBD/MI what is usually called "initial activation" versus "auto-amplification" components required for switch/like maximal MPF activity. The authors could have reported and discussed the dual ARPP19 phosphorylations by Greatwall and CDK1.

This is a great point. We have now included an additional discussion of the S69 Cdk1 and S165 PKA sites on ARPP19 in manuscript (see Figure 3-S5). Briefly, we find that the phosphorylation sites S106 (Greatwall) and S165 (PKA) display similar temporal behavior, whereas S69 (Cdk1) phosphorylation increase at the Prophase 1 to GVBD transition, remain high during MI to 2-PN, and further increase during first cleavage.

3. Page 7. The authors use morpholino oligonucleotides to block the translation of cyclin A and B. However, they provide no control data on the extent and selectivity of the knockdown. This control is necessary to draw any conclusion.

Thank you for this suggestion. We have now included a new experiment with Western blotting using Cyclin B antibodies (a gift from the Kishimoto lab) for first cleavage embryos following injection of either control, Cyclin A, or Cyclin B morpholinos. We find that Cyclin B protein is robustly and specifically depleted following Cyclin B morpholino injection. Unfortunately, we lack antibodies against starfish Cyclin A. In addition, as these are start site-targeting morpholinos, they have no measurable effect on mRNA levels that could be assessed by RT-PCR. Therefore, to increase confidence in the Cyclin A depletion phenotype, we have also included a Western blot for TPxK phosphorylation. As predicted, individual depletion of either Cyclin A or Cyclin B results in a decrease in phosphorylation levels relative to controls. We believe that these added experiments, combined with the robust and specific phenotypes that are consistent with prior expectations for the roles of these cyclins during meiotic progression, provide high confidence in our morpholino results. These results have been added to Figure 2—Figure Supplement 2.

4. Results, page 13. In the revised manuscript, the authors have included additional data with a more informative sampling rate between MI and MII to include points at anaphase. Using TP and SP specific antibodies the authors confirm that dephosphorylation at the TP site has faster kinetics than dephosphorylation at SP sites. The authors conclude that "despite the transient inactivation of Cdk1 between the meiotic division, SP phosphorylation remains relatively stable during this window". They use this finding to rule out the possibility of a kinase rephosphorylating at the SP sites. However, the authors do not discuss the possibility that the divergent MAPK activity, which remains constantly high, while Cdk1 activity is declining, at least in part contributes to divergent SP/TP phosphorylation state. This should be further discussed and possibly addressed experimentally.

We agree with this point. We have now added a statement to the Discussion section stating that MAPK activity remains high until the MII to 2PN transition and could be responsible for continuous phosphorylation of proline-directed serine and threonine phosphorylation. However, even if this is the case, the presence of a phosphatase activity with preferential threonine dephosphorylation activity is still needed to explain our results for differential dephosphorylation, since MAPKs do phosphorylate both serine and threonine residues.

5. To disrupt PP2A-B55 function, the authors use a dominant-negative approach with point mutations in B55 that disrupt interactions with substrates. The authors provide some data on the effect on meiotic progression. However, they do not investigate whether disruption of PP2A-B55 anchoring indeed blocks the divergent TP/SP dephosphorylations. This could be explored using phospho-specific antibodies.6. The authors impute the divergent dephosphorylation of TP and SP sites to the selectivity of PP2A-B55. If this were correct, one should find a mirror image of dephosphorylation observed at the MI/MII transition during prophase to GVBD transition. PP2A is active in prophase 1, but it becomes inactivated at GVBD/MII. One should then see a preferential dephosphorylation of TP sites in prophase with little change in the SP sites. The authors should discuss this possibility and provide the data they have available; a hint of this is reported in Figure 6A.

This is a fantastic suggestion. For the revised manuscript, we have compared the abundance of single, localized, and reproducibly-quantified phosphopeptides in the dataset for TP and SP consensus motifs. We found that the average abundance of SP phosphorylation sites in Prophase I was 0.3 compared to 0.14 for TP sites (p-value <0.0001). This indicates that TP sites are more readily opposed in Prophase I. Intriguingly, this trend is reversed in GVBD, where the average abundance of SP phosphorylation sites is Prophase I was 0.66 compared to 0.74 for TP sites (p-value <0.0001). We have added the data showing the relative SP vs. TP phosphorylation abundance in Prophase I to Figure 4-S1.

7. INCEMP is a highly phosphorylated protein with a large number of sites to choose from. The authors report changes in the TP and SP sites, which clearly diverge. This is considered a major strength of the study, as it verifies that spatial sequestration of the protein is not an issue in the differential TP/SP dephosphorylation. The authors should include the SP phosphorylation patterns also for PRC1 and TPX2. Now only TP patterns are reported in Figure 5-S3. This would confirm that the TP/SP divergent dephosphorylation is of wide applicability.

We appreciate this helpful suggestion. We have now added this data to Figure 5-S3.

8. In the discussion the authors make no attempt to compare and contrast their data with the wealth of information reported for other species. The authors do not specify the species used in the title. In the same vein, in the opening statement of the discussion the authors claim to have defined an extensive program of phosphorylation during the oocyte-to-embryo transition. However, they do not mention that they do so in the sea star. This has already been reported for other species. In addition, the authors do not discuss that the sea star provides a simplified experimental model to investigate regulation of meiosis and that many additional layers of regulation are present in frogs and mammals. Thus, the mechanisms of meiotic resumption vary considerably among species including sea star, frog and mouse. This important concept is not captured in the discussion.

We now added a more detailed comparison of our findings to those in other organisms to the discussion.